# PaleoSTeHM v1.0: a modern, scalable spatio-temporal hierarchical modeling framework for paleo-environmental data

Yucheng Lin[1], Robert E. Kopp[1,2], Alexander Reedy[1,2], Matteo Turilli[3,4], Shantenu Jha[2,3,5], and Erica L. Ashe[1]

[1]Department of Earth & Planetary Sciences, Rutgers University, Piscataway, NJ, USA
[2]Rutgers Climate and Energy Institute, Rutgers University, New Brunswick, NJ, USA
[3]Department of Electrical and Computer Engineering, Rutgers University, Piscataway, NJ, USA
[4]Computational Science Initiative, Brookhaven National Laboratory, Upton, NY, USA
[5]Computational Science Department, Princeton Plasma Physics Laboratory, Princeton, NJ, USA

**Correspondence:** Yucheng Lin (yc.lin@rutgers.edu) and Robert E. Kopp (robert.kopp@rutgers.edu)

**Abstract.** Geological records of past environmental change provide crucial insights into long-term climate variability, trends, non-stationarity, and nonlinear feedback mechanisms. However, reconstructing spatio-temporal fields from these records is statistically challenging due to their sparse, indirect and noisy nature. Here, we present PaleoSTeHM, a scalable and modern framework for spatio-temporal hierarchical modeling of paleo-environmental data. This framework enables the implementation of flexible statistical models that rigorously quantify spatial and temporal variability from geological data while clearly distinguishing measurement and inferential uncertainty from process variability. We illustrate its application by reconstructing temporal and spatio-temporal paleo sea-level changes across multiple locations. Using various modeling and analysis choices, PaleoSTeHM demonstrates the impact of different methods on inference results and computational efficiency. Our results highlight the critical role of model selection in addressing specific paleo-environmental questions, showcasing the PaleoSTeHM framework's potential to enhance the robustness and transparency of paleo-environmental reconstructions.

## 1 Introduction

As humans push the planet's climate and biosphere increasingly far outside the range of our species' experience, the environmental reconstructions derived from the geological record provide critical out-of-sample information to test the physical models used to project future environmental change. However, as environmental records, geological data are sparse, often noisy, and indirect (PAGES2k Consortium, 2017; Shennan, 2015). Reconstructing paleo-environmental fields is thus a critical and challenging statistical task (Tingley et al., 2012).

From a modeling perspective, spatio-temporal hierarchical statistical models provide a natural, conceptually straightforward framework for reconstructing paleo-environmental signals (Ashe et al., 2019; Cressie and Wikle, 2015; Tingley et al., 2012). Hierarchical statistical models, often employed within a Bayesian framework, decompose the various sources of random variation contributing to individual observations into distinct levels, thereby providing a clear articulation of the assumptions underlying the statistical analysis. They have been increasingly used to model paleo-climate fields from geological proxies,

which are naturally occurring physical characteristics or chemical markers that can be used to reconstruct past climate and environmental conditions, such as temperature and precipitation, from sources like tree rings and corals (Walter et al., 2022; PAGES2k Consortium, 2017). These applications have proven crucial in assessing the robustness of scientific knowledge of past climate and placing changes in the modern, instrumentally observed period in the context of longer-term variability. For example, they have shown an increasing influence of ice melt and thermal expansion on global mean sea level (GMSL) since 1860 CE (Walker et al., 2021), that GMSL rise over the 20th century was faster than during any century in at least 3000 years (Kemp et al., 2018; Kopp et al., 2016) and that several early 21st century Arctic summers exhibited warmth unprecedented in at least 600 years (Tingley and Huybers, 2013).

*Hierarchical models* are in high demand within the paleoenvironmental research community. For example, in the past few years, numerous papers have used temporal or spatio-temporal hierarchical models with *Gaussian Process (GP)* priors to interpret paleo sea-level proxies (e.g., Tan et al., 2023; Khan et al., 2022; Vacchi et al., 2021). To meet the demand of the paleo-environmental research community, this paper describes PaleoSTeHM v1.0, which is designed to support the flexible and high-performance implementation of spatio-temporal hierarchical modeling for paleo-environmental data. PaleoSTeHM (https://github.com/radical-collaboration/PaleoSTeHM) is a framework built in the spirit of open science and utilizes modern machine learning architecture (e.g., Pollack et al., 2024). It is designed so users can select not only various modeling choices, such as change-point models for temporal analysis or GP for spatio-temporal analysis, but also *analysis choices*, including *fully Bayesian*, *empirical Bayesian* and *variational Bayesian analysis* (more details in section 2), to investigate different research questions, with different types of data and spatio-temporal scales (e.g., local to global, years to millennia) considered. In this paper, some key terms and phrases are defined in Table 1 and are italicized upon their first occurrence for clarity.

## 2 Hierarchical statistical modeling

Hierarchical modeling is a statistical approach that separates multiple sources of variability contributing to individual observations into distinct levels, enabling a clear understanding and quantification of uncertainties. This section briefly describes basic theory of hierarchical modeling in the paleo-environment, using paleo sea level as an illustrative example. For more systematic introductions to hierarchical statistical modeling of paleo sea-level and paleo-climate, readers can refer to Ashe et al. (2019) and Tingley et al. (2012).

*Bayesian statistics* denotes a statistical theory that uses Bayes' theorem to update probabilities conditioned on data and prior knowledge. Based on Bayes' theorem (Laplace, 1810), the *conditional probability* of the observed data ($y$) can be derived from the conditional probability of unknown *parameter(s)* or process(es) ($\theta$):

$$p(\theta|y) = \frac{p(y|\theta)p(\theta)}{p(y)} \tag{1}$$

where $p$ denotes 'probability' and | represents 'given'. The *likelihood* function, $p(y|\theta)$ represents the probability of observing the data $y$ given the parameter(s) or process(es) $\theta$ of the model. The *prior distribution*, $p(\theta)$, captures *a priori* beliefs about

**Table 1.** Definitions of relevant terms in this study. This paper employs terminology based on Ashe et al. (2019).

| Term | Meaning |
| --- | --- |
| analysis choices | decisions in how to implement a specific model structure, including the selection of deterministic or probabilistic methods (e.g., Bayesian analysis) and approaches to incorporate temporal uncertainty, such as errors-in-variables frameworks or noisy-input methods |
| auto-differentiation | automatic differentiation, a set of computational techniques to automatically evaluate the partial derivative of a function |
| Bayesian statistics | a statistical theory that uses Bayes' theorem to update probabilities conditioned on data and prior knowledge |
| conditional probability | a measure of the probability of an event occurring, given that another event is already known to have occurred |
| continuous core | near-continuous records from a single core of sediment or a single coral reef |
| covariance function | defines prior beliefs about the relationship or correlation between variables |
| data level | model representations of the relationship between the phenomenon and observed data |
| errors-in-variable (EIV) | a fully-Bayesian framework that accounts for measurement uncertainty in independent variables |
| error | difference between a measurement and the true value |
| empirical Bayesian analysis | an analysis choice to estimate the prior from data by fixing higher-level parameters at their most likely values, typically determined using maximum likelihood estimation |
| fully Bayesian analysis | an analysis choice that assigns prior distributions to all model parameters, combining prior knowledge and observed data to shape the posterior distribution. Since the posterior is often analytically intractable, MCMC methods are used to approximate it |
| Gaussian Process (GP) | a stochastic process that generalizes the multivariate Gaussian distribution to continuous time and space, defined by mean and covariance functions |
| GPU acceleration | a computational technique that leverages Graphics Processing Unit (GPU) to significantly accelerate machine learning computational processes |
| glacial isostatic adjustment (GIA) | Earth surface water mass redistribution induced solid Earth deformation and gravitational and rotational fields variation |
| hierarchical model | a statistical framework that partitions the multiple random effects that lead to individual observations into different levels |
| hyperparameter | parameter of a prior distribution |
| isotropy | a property of having identical statistical characteristics in all directions |
| likelihood | the probability of observing the given data as a function of the model parameters |
| Markov Chain Monte Carlo (MCMC) | techniques used to generate random variables, perform complicated calculations and simulate complicated distributions through random sampling in Bayesian models |
| noisy-input | a framework that applies a first-order Taylor series approximation to account for errors in the independent variable (e.g., time) thereby translating these into equivalent errors in the dependent variable |
| non-parametric model | a model that does not involve any assumptions about the functional form of the relationship between variables |
| optimization | the process of iteratively improving the accuracy of a machine learning model, lowering the degree of error |
| parameter | a quantity used in mathematical equations, computer programs, physical models, and other scientific applications to describe the characteristics of a system or process |
| parameter level | model representations of prior beliefs about parameters used to control the behavior of a statistical and/or physical model at different levels of the hierarchy |
| physical model | a class of model based on physical principles to describe natural phenomena, typically using mathematical representations of a system or process that uses numbers and equations to describe physical conditions |
| probabilistic programming | a programming paradigm that integrates probabilistic models and inference algorithms into standard programming languages, enabling users to define complex statistical models and automatically perform inference on uncertain variables |
| parametric model | a model that explicitly assumes a specific functional form or mathematical relationship between variables, defined by a fixed set of parameters that summarize the underlying process |
| process level | model representations of the underlying processes responsible for the data generation |
| posterior distribution | a type of conditional probability that results from updating the prior probability with observational information summarized by the likelihood |
| prior distribution | the assumed probability distribution before any observational evidence is taken into account, which can be uninformative or subjective based on a priori knowledge |
| residual | the difference between an observed and a modeled or predicted value |
| relative sea level (RSL) | vertical distance between the solid Earth surface and the ocean surface. In the paleo sea-level context, RSL is typically measured relative to present day, where a positive value indicates a higher RSL and a negative value indicates a lower RSL |
| smoothness | the characteristic of a process that reflects the gradualness of its variations over time or space, often controlled by the kernel's differentiability in Gaussian Process models |
| sampling covariance function | a function that describes the covariance structure of a random process, derived from the variability observed in a set of model ensembles or sampled data |
| space-time separability | a property of processes where the spatial and temporal components of the covariance function are treated as independent, so the covariance is expressed as a product of purely spatial and purely temporal functions |
| stationarity | a property of processes or signals where their statistical properties, like mean and variance, remain consistent over time or space |
| uncertainty | a parameter defining the range within which a measured value is likely to fall, given a specified probability. |
| teleconnection | a causal connection or correlation between meteorological or other environmental phenomena which occur a long distance apart |
| variational Bayesian analysis | an analysis choice that approximates the full posterior distribution with a simpler parametric distribution, transforming Bayesian inference into an optimization problem to reduce computational expense while estimating uncertainty |
| white noise | serially uncorrelated random variation (zero mean and finite variance) |

the unknown parameter(s) or process(es) before any data are observed. The term $p(y)$, known as the marginal likelihood (or evidence), is the probability of the observed data averaged over all possible parameters or processes:

$$p(y) = \int p(y \mid \theta) p(\theta) \, d\theta \tag{2}$$

Given the observations, the *posterior distribution*, $p(\theta|y)$, reflects the updated beliefs about the parameter(s) or process(es) after considering the data. Since the marginal likelihood $p(y)$ is often intractable and remains constant for a given data set, we use the simplified form of Bayes' theorem, where the posterior distribution is proportional to the product of the likelihood and the prior:

$$p(\theta|y) \propto p(y|\theta) p(\theta) \tag{3}$$

where $\propto$ indicates 'is proportional to.'

A basic hierarchical statistical model distinguishes the change in observations from both its inherent variability and the observational noise. These models achieve probabilistic uncertainty estimation for time series and/or spatial fields by treating observed data as conditional on a latent (unobserved) process and unknown parameters, enabling separate quantification of

65 uncertainties at each level through the application of Bayesian conditional probabilities. Each level of the model quantifies uncertainties independently, necessitating careful evaluation of their respective sources. Generally, three levels are defined, the *data level*, the *process level* and the *parameter level*:

$$p(f, \theta_s, \theta_d \mid y) \propto \underbrace{p(y \mid f, \theta_d)}_{\text{data model}} \cdot \underbrace{p(f \mid \theta_s)}_{\text{process model}} \cdot \underbrace{p(\theta_d, \theta_s)}_{\text{parameter model}} \tag{4}$$

Taking paleo *relative sea-level (RSL)* change as an example, the data level model defines the relationship between the latent

(unobserved) RSL process ($f$) and the observed RSL data (instrumental and/or proxy), $y$, while accounting for measurement, inferential (e.g., arising from converting a proxy's elevation to a distribution of RSL) and dating uncertainties (often inherited from geochronology techniques, Reimer et al., 2020; Wright et al., 2017). This level represents the probability distribution of observing a particular sea-level height at a given age, conditioned on the underlying latent process and the associated uncertainties, encapsulated by the data level parameters, $\theta_d$.

The process level distinguishes the underlying phenomenon of interest and its inherent variability, from the noisy observation captured at the data level. This model integrates scientific understanding and associated uncertainties into the estimation of the true RSL process conditioned on model parameters, $\theta_s$. These parameters may represent unobserved *physical model* parameters (e.g., Earth's rheology in a *glacial isostatic adjustment (GIA)* model), statistical model parameters (such as the rate of change in a sea-level model), or *hyperparameters* (parameters of a prior distribution, such as length scale and variance in a GP model).

At the foundational level, the parameter model specifies the prior distribution for all unknown parameters, effectively capturing the essential characteristics of both the data and process levels through the unobserved parameters.

In addition to constructing models at the data, process and parameter levels, often referred to as modeling choices (Ashe et al., 2019), it is essential to choose an appropriate analysis choice for a specific model. This involves decisions regarding

the implementation of a model structure, such as deterministic methods or probabilistic methods like Bayesian analysis. Deterministic methods, such as least-squares analysis (Wilks, 1938) and likelihood maximization (Aitken, 1936), rely on fixed relationships between states and events without incorporating randomness into the modeling process. In contrast, probabilistic methods, like Bayesian analysis (Hastings, 1970), account for uncertainty explicitly by representing model parameters and outputs as probability distributions, enabling flexible and robust uncertainty quantification. Analysis choices are also integral to addressing how measurement uncertainties, particularly those arising from geochronological techniques (i.e., input uncertainty), are incorporated and managed within the model. This ensures that the uncertainty is properly quantified and reflected in the final analysis outputs (Ashe et al., 2019). The selection of modeling and analytical choices should consider the problem's complexity, data size and resolution, computational resources, and prior knowledge.

## 3 Software architecture

This section provides a comprehensive overview of PaleoSTeHM, detailing its foundational model implementation (section 3.1), the basic architecture for a typical PaleoSTeHM experiment (section 3.2) and the development of PaleoSTeHM modules (sections 3.3, 3.4 and 3.5).

### 3.1 Model implementation

PaleoSTeHM is designed to be a functionally extensible and high-performing toolkit for modeling paleo data. It is fully open-source and developed under a four-layer structure to maintain a flexible and generic design that is agile to future development (Figure 1). The four layers, shown from the bottom to top in Figure 1) are: (L1) computing platforms, (L2) machine learning platforms, (L3) PaleoSTeHM modules and (L4) PaleoSTeHM users. At the fundamental level, PaleoSTeHM utilizes computational power from various platforms (L1), such as clouds, clusters and high-performance computing systems, to ensure scalability and flexibility for diverse applications. Built upon L1, L2 employs Python as the user interface language, leveraging PyTorch (Paszke et al., 2019) and Pyro (Bingham et al., 2019) as its high-performance machine learning platforms. The fast-evolving ecosystem of these popular machine learning platforms enables PaleoSTeHM to support *probabilistic programming*, *auto-differentiation*, *GPU acceleration*, and state-of-the-art *optimization* algorithms, making it highly efficient and adaptable to a variety of paleoenvironmental statistical tasks.

The core toolkit and development reside in L3, which comprises modules that integrate existing machine learning capabilities from L2. This layer includes three primary components: (1) the Modeling Choices module, which provides options for data, process, and parameter-level modeling (section 3.3); (2) the Gaussian Process kernel module, a sub-module of the Modeling Choices module that supports kernel construction using GP priors (section 3.4); and (3) the Analysis Choices module, which incorporates methods to propagate temporal uncertainty into inference results (i.e., temporal uncertainty treatment) and Bayesian inference (section 3.5). These modules enable flexible and efficient spatio-temporal hierarchical modeling for a wide range of paleo-environmental applications.

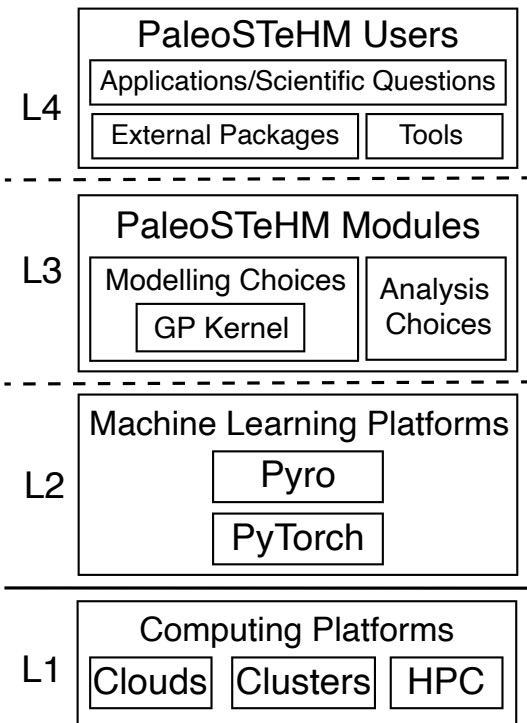

**Figure 1.** Schematic illustration of the four-layer structure of PaleoSTeHM. L1 specifies various computing platforms (clouds, clusters and HPC), L2 comprises machine learning platforms (Pyro, Bingham et al., 2019, and PyTorch, Paszke et al., 2019), L3 includes PaleoSTeHM modules (Modeling Choices, GP kernel and Analysis Choices, see Figure 2) and L4 consists of the user layer, facilitating interaction with external packages and tools for practical applications and scientific inquiries.

We anticipate PaleoSTeHM interacting with external packages and/or tools for practical applications and addressing scientific questions on the PaleoSTeHM User layer (L4, Figure 1). Here, 'External Packages' refer to external Python libraries, which provide various pre-processing and post-processing data functions. For example, in PaleoSTeHM tutorials (see section 4), we use Scipy (Virtanen et al., 2020) for interpolation and Matplotlib (Hunter, 2007) for visualization. 'Tools' represent frameworks and services adapted by other developers to integrate PaleoSTeHM capabilities into their toolkits (e.g., Frame-

work for Assessing Changes To Sea level (FACTS); Kopp et al., 2023). Such plug-in implementations will make it easy for users drawn from any of the PaleoSTeHM categories to use, extend, or contribute to core capabilities for various scientific applications.

## 3.2   PaleoSTeHM modeling workflow

Constructing and optimizing a hierarchical model within PaleoSTeHM involves a workflow consisting of five sequential selec-

125 tion steps (outlined in Figure 2), with a focus on modeling and analysis choices in layer L3 as depicted in Figure 1. Typical

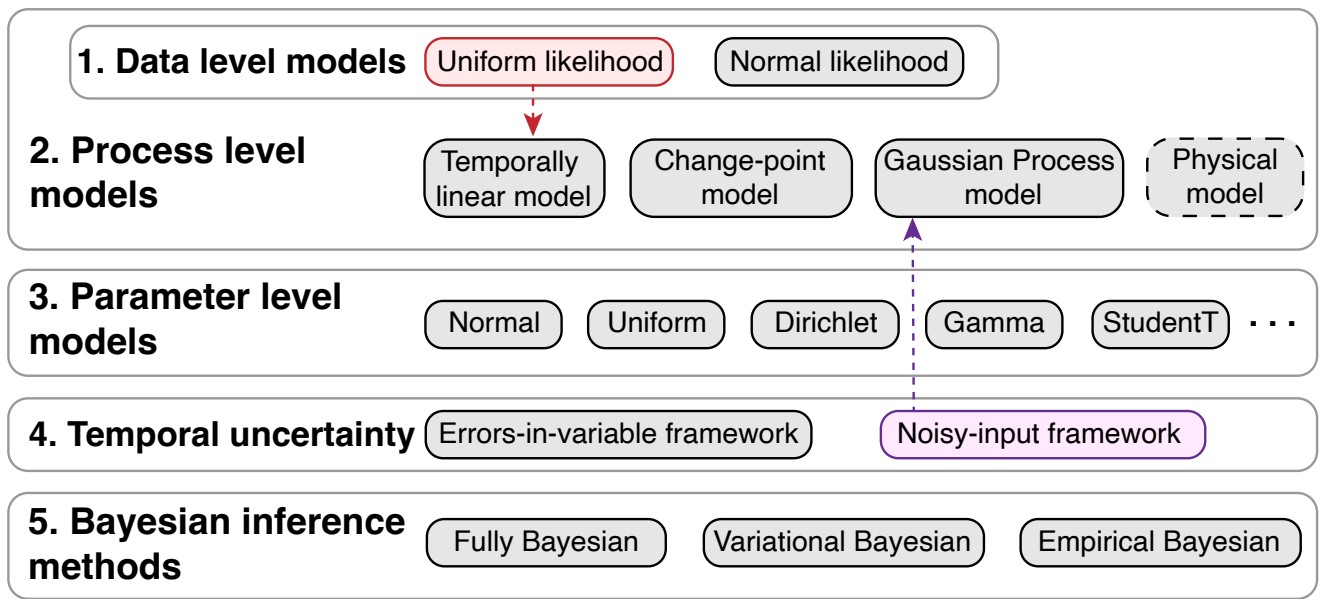

**Figure 2.** A schematic illustration of the PaleoSTeHM modeling workflow, providing more detailed information about layer L3 in Figure 1. The large numbered boxes represent five steps to build a hierarchical model and it should be noted that the data-level model is specified within each process-level model in PaleoSTeHM v1.0. The smaller boxes indicate different modeling choices within each step. Grey boxes denote available choices that apply to other grey boxes in different steps. Red and purple boxes represent a specific data level model and temporal uncertainty treatment method corresponding to a specific process level model (e.g., temporally linear and Gaussian Process models), as indicated by colored arrows. The dashed grey box (physical model) highlights that no specific physical model is implemented in PaleoSTeHM. Instead, PaleoSTeHM utilizes outputs from other physical models (see section 3.3.2.)

PaleoSTeHM experiment steps include: (1) selecting data-level models for paleo-environmental data; (2) choosing an appropriate process-level model to describe the latent process; (3) defining prior distributions for each model parameter; (4) selecting a temporal uncertainty treatment method; and (5) choosing a Bayesian inference method (Figure 2). These five steps reflect core functionalities of PaleoSTeHM Modules (layer L3 shown in Figure 1). To support the effective selection of modeling

and analytical choices provided by PaleoSTeHM for various paleo-environmental applications, the fundamental theories and example applications for each modeling choices will be introduced in section 3.3.

### 3.3 Modeling Choices module

As mentioned above, spatio-temporal hierarchical modeling experiments begin with selecting an appropriate modeling choice for a specific problem. This module offers a variety of commonly used temporal and spatio-temporal modeling options for

paleo-environmental studies (Figure 2). While this paper does not include a dedicated section for parameter-level modeling, the integration of Pyro (Bingham et al., 2019) and PyTorch (Paszke et al., 2019) enables users to define prior probabilities for

data and process-level model parameters using a wide range of commonly used probability distributions. This functionality allows users to customize priors as needed for their specific modeling requirements.

### 3.3.1 Data level modeling

The data level of a hierarchical statistical model characterizes the relationship between true (unobserved) target signals and uncertain observations due to multiple *error* sources. For example, in reconstructing past sea-level changes, the data level addresses uncertainties arising from elevation measurements, indicative range and leveling errors (Khan et al., 2017). Additionally, proxy data are often subject to inherent temporal uncertainties stemming from various geochronological methods (e.g., radiocarbon dating, Reimer et al., 2020; Heaton et al., 2020). This relationship between observed data and latent process can

be formally expressed as:

$$y_i = f(x_i, t_i) + \epsilon_i^y \tag{5}$$

$$t_i = \hat{t}_i + \epsilon_i^t \tag{6}$$

where $y_i$ is the observed data, $x_i$ is the noise-free spatial location of $i$-th observation, $t_i$ is its true age, $\hat{t}_i$ is the mean obser-

150 vational age and $\epsilon_i^t$ and $\epsilon_i^y$ are uncertainties in the age measurement and target signal reconstruction. For paleo-environmental studies, a commonly made assumption is that both $\epsilon_i^t$ and $\epsilon_i^y$ are multivariate normally distributed with zero mean and heteroscedastic covariance, so $\epsilon^y$ can be expressed as:

$$\epsilon^y \sim \mathcal{N}(0, \Sigma_y) \tag{7}$$

$$\Sigma_y = \begin{pmatrix} var(y_1) & cov(y_1, y_2) & \cdots & cov(y_1, y_n) \\ cov(y_2, y_1) & var(y_2) & \cdots & cov(y_2, y_n) \\ \vdots & \vdots & \ddots & \vdots \\ cov(y_n, y_1) & cov(y_n, y_2) & \cdots & var(y_n) \end{pmatrix} \tag{8}$$

where $n$ indicates the number of observations available, $var(\cdot)$ represents the variance of specific data and $cov(\cdot, \cdot)$ stands for covariance between two data points, which is often assumed to be 0 when all data are assumed to be independently distributed. However, in practice, strong correlations in paleo-environmental data can emerge from shared processes or dependencies, such as sedimentary records from the same core or data dated using age-depth modeling techniques, where shared depositional

history introduces correlated uncertainties (Cahill et al., 2015; Blaauw, 2010). Ignoring these correlations can lead to biased estimates and reduced model reliability. Adapting the likelihood structure to account for covariance, for example, by using a structured covariance model from age-depth modeling, allows for more accurate and robust inference (Cahill et al., 2015).

In PaleoSTeHM v1.0, the data-level model is specified within each process-level model, which is assumed to be normally and independently distributed (Figure 2). For illustrative purposes, PaleoSTeHM v1.0 also includes an implementation of

uniform likelihood together with a temporally linear model (see Figure 2 and section 4.1). For specific problems requiring different likelihood structures, users can replace the likelihood sampling code (a probabilistic random sampling operation in Pyro) to utilize most of the standard probability distributions supported by Pyro, such as multivariate Normal distributions with covariance structures mentioned above.

### 3.3.2 Process level modeling

The process level is a hierarchical layer where the variability of the paleo-environment signal is modeled and, in certain cases, decomposed. The process level reflects a scientific understanding of environmental change processes. PaleoSTeHM v1.0 offers multiple process-level models for temporal or spatio-temporal data analysis.

**Temporally linear models.** Starting with temporal data analysis, probably the most straightforward method for estimating linear trends and the average rate of paleo-environmental change is to fit a linear model to the observed data over time (i.e., straight line model). For example, Engelhart et al. (2009) and Islam et al. (2021) applied linear regression to discrete paleoenvironment data to estimate respectively the average rate of sea-level, rainfall and temperature change during specific time intervals. Over those periods, the observations were assumed to be well represented by a linear trend. A temporally linear model can be expressed as:

$$f(t) = \alpha + \beta t \tag{9}$$

where $f(t)$ is the modeled true RSL, $\beta$ is the constant rate of change in paleo-environmental variable and $\alpha$ is the intercept (Ashe et al., 2019).

**Change-point models.** Change-point models describe a single time series by partitioning it into distinct, contiguous segments, each characterized by a linear trend over time (Carlin et al., 1992). These models are widely used to identify the timing of abrupt changes in past climate conditions. For instance, Caesar et al. (2021) and Kemp et al. (2015) respectively employed change-point models to determine the onset of reduced strength in the Atlantic Meridional Overturning Circulation and the commencement of modern sea-level rise in Connecticut. With $m$ change points, the change-point model can be written as:

$$f(t) = \begin{cases} \alpha_1 + \beta_1(t - \gamma_1), & \text{when } t < \gamma_1 \\ \alpha_{j-1} + \beta_j(t - \gamma_{j-1}), & \text{when } \gamma_{j-1} < t < \gamma_j \\ \alpha_m + \beta_{m-1}(t - \gamma_m), & \text{when } \gamma_m \leq t \end{cases} \tag{10}$$

where $\gamma_k$ represents a change point, $\alpha_k$ denotes the expected value of RSL at that change point and $\beta_j$ indicates the rate of RSL change for each of the $m+1$ segments. This model incorporates a continuity constraint ensuring that $\alpha_k$ equals $\alpha_{k-1}$ plus the product of $\beta_{k-1}$ and the difference between $\gamma_k$ and $\gamma_{k-1}$. In PaleoSTeHM, the change-point model is implemented to allow users to specify any number of change points (i.e., $m$ in equation 10) in the model.

**Gaussian Process models.** GP modeling is a *non-parametric* Bayesian approach that has been frequently used to infer temporal (or spatio-temporal) variation of paleo-environmental change, including magnitude and rate (Ashe et al., 2019). In

models with GP priors, the relationships among any set of points (e.g., over time or across both space and time) are described
by a multivariate normal distribution, fully characterized by a mean function and a *covariance function* (or kernel). Unlike
*parametric models*, such as linear or change-point models used for spatio-temporal analysis, GP models offer greater flexibility
because the shape of the curve is determined by the covariance matrix, which reflects the relationship between data points and
is inferred directly from the data, rather than being restricted by a predefined functional form.

GP models have gained considerable traction in paleo-environmental science, largely owing to their proficiency in extracting
meaningful insights from relatively small datasets. They utilize a non-parametric framework to interpret intricate data patterns
effectively. For example, Kay et al. (2021) utilized a GP model to assess herbivore richness for different latitudes in Argentina.
Apart from that, Walker et al. (2021) estimated the trend and rate of RSL change across the US Atlantic coast with a GP model.
A spatio-temporal GP model, which is defined by its mean function, $\mu(X)$ and covariance function (i.e., kernel) $K(X, X')$,
can be expressed as:

$$f(X) \sim GP(\mu(X), K(X, X')) \tag{11}$$

where $X$ indicates spatio-temporal location. A popular choice for many paleo-environmental studies is using the zero-mean
function, indicating $\mu(X) = 0$ everywhere. In this case, the predictions are only determined by covariance function $K(X, X')$,
which defines prior expectations about how information is shared between points in different time and space, which typically
decays as the time and space differences increase (Rasmussen and Williams, 2006).

Constructing the covariance function is a pivotal and challenging step in a GP model, as it significantly influences the out-
come of the inference results. Yet, justifying the form of the covariance function in Gaussian processes for paleoenvironmental
studies can be challenging because the processes being modeled are influenced by a wide range of spatial and temporal de-
pendencies, many of which are complex, nonstationary, and not well understood (Tingley et al., 2012; Stein, 2012, 2005a).
PaleoSTeHM addresses this by incorporating a 'GP kernel' module under the Modeling Choices module, designed to offer
more flexibility and customization extendability. This module provides a user-friendly platform for creating and managing GP
kernels, streamlining the process of model construction and enhancing the adaptability of the analysis to diverse problems. For
paleo-environmental applications, multiple choices of building kernels have been adopted in various studies (e.g., Walker et al.,
2021; Hay et al., 2015; Kopp et al., 2016, 2014, 2009), and some examples are shown in section 4.2.

**Physical models.** A physics-based model simulates real-world changes with predictive capabilities anchored in the causal
mechanisms delineated by the laws of physics (Saltzman, 2001; Farrell and Clark, 1976). Comparatively, statistical models
mostly depend on data-driven correlations, often overlooking fundamental physical principles (e.g., mass or energy conser-
vation). Examples in paleo-environment research include using global circulation models to understand the response of the
climate system to different climate forcings (Kageyama et al., 2018) and employing ice sheet dynamic models to quantify past
ice sheet response to climate change (DeConto and Pollard, 2016; Tarasov et al., 2012). In the realm of paleo sea-level change
modeling, the GIA model is a widely adopted tool to characterize sea-level changes driven by the gravitational, rotational
and deformational (GRD) effects resulting from the redistribution of ice and water mass (e.g., Lin et al., 2023a; Whitehouse,
2018). The predictive power of such a model is contingent upon underlying formulation and core physical parameters (Peltier

et al., 2015; Kendall et al., 2005; Peltier, 2004), such as the history of ice sheet fluctuations and the rheological properties of the Earth's interior for a GIA model (Lin and Yousefi, 2025; Austermann et al., 2013). Validating the physical model against observational data should allow a more accurate representation of spatially dependent patterns of sea-level change, including those linked to sea-level fingerprints (Lin et al., 2021; Mitrovica et al., 2001), in stark contrast to statistical models that might merely presume correlation diminishes with distance (Walker et al., 2021).

Although PaleoSTeHM does not include a specific type of physics-based model (Figure 2), it offers multiple options to incorporate physical model outputs into final estimates (see examples in section 4.2). Users can use PaleoSTeHM to probabilistically calibrate physical model ensembles conditioned upon observational data. For instance, latent paleoenvironmental processes can be modeled as a combination of physical model ensembles conditioned on different physical parameter combinations, using a Dirichlet distribution prior. PaleoSTeHM also supports using a physical model as a mean function in a GP model. In this context, the GP covariance function essentially models the *residuals*—those processes not captured by the physical model—between observations and the predicted mean function. Additionally, PaleoSTeHM facilitates the construction of *sampling covariance functions* derived from a physical model ensemble, further enhancing its utility in model integration and assessment (Hay et al., 2015). All of these capabilities are demonstrated in section 4.2, with accompanying source code provided in the PaleoSTeHM GitHub page (see code and data availability).

## 3.4 Gaussian Process Kernel module

The GP Kernel module in PaleoSTeHM is a cornerstone for modeling spatial and temporal variations in paleo-environmental data based on GP priors (Figures 1 and 2). It includes a variety of widely used kernels as described in Rasmussen and Williams (2006). In paleoenvironmental studies, examples of kernel applications include the linear (or dot-product) kernel (Khan et al., 2017), radial basis function kernel (Cahill et al., 2015), rational quadratic kernel (Turner et al., 2023; Hay et al., 2015), Matérn kernel (Walker et al., 2021; Kopp et al., 2016), and periodic kernel (Meltzner et al., 2017). These kernels characterize features such as *stationarity*, *isotropy*, *smoothness* and periodicity in Gaussian processes (Ashe et al., 2019). Detailed kernel information is given in Table 2.

Each kernel possesses unique characteristics and necessitates specific parameters (Table 2). For instance, the linear kernel produces linear trends identical to a temporally linear model, suitable for modeling signals with long temporal length scales (e.g., tectonic and GIA in Common Era and future sea level modeling; Kopp et al., 2016, 2014). The radial basis kernel and the Matérn family of kernels are highly generalizable and allow specification of the degree of differentiability (Table 2), making them suitable for representing physical processes with different levels of smoothness. For example, the GRD effects related to GIA are spatio-temporally smooth, while sediment compaction-induced sea-level rise can be much more localized and rough (i.e., less differentiable, Kopp et al., 2016; Mitrovica et al., 2011).

In the GP Kernel module of PaleoSTeHM v1.0, all kernels are designed for process-level modeling to capture temporal and/or spatial correlations, except for the temporal and spatial *white noise* kernels, which account for additional measurement errors or unstructured variability by introducing serially uncorrelated uncertainty at the data level (equation 5). Apart from the linear kernel, all included kernels are stationary and isotropic (Table 1). To enhance kernel construction flexibility, PaleoSTeHM

**Table 2.** Summary of Gaussian Process kernels in PaleoSTeHM, based on Stein (2012) and Rasmussen and Williams (2006).

| Kernel Name | Supports Spatial Data[*] | Differentiability | Equation † |
|---|---|---|---|
| Radial Basis Function | Yes | Infinitely differentiable | $k(X, X') = \sigma^2 \exp\left(-\frac{1}{2}\frac{\|X-X'\|^2}{\ell^2}\right)$ |
| Rational Quadratic | Yes | Infinitely differentiable | $k(X, X') = \sigma^2 \left(1 + \frac{\|X-X'\|^2}{2\alpha\ell^2}\right)^{-\alpha}$ |
| Periodic | No | Infinitely differentiable | $k(X, X') = \sigma^2 \exp\left(-2\frac{\sin^2(\pi(X-X')/p)}{\ell^2}\right)$ |
| 2/1 Matérn | Yes | Non-differentiable | $k(X, X') = \sigma^2 \exp\left(-\frac{\|X-X'\|}{\ell}\right)$ |
| 3/2 Matérn | Yes | Once differentiable | $k(X, X') = \sigma^2 \left(1 + \sqrt{3}\frac{\|X-X'\|}{\ell}\right) \exp\left(-\sqrt{3}\frac{\|X-X'\|}{\ell}\right)$ |
| 5/2 Matérn | Yes | Twice differentiable | $k(X, X') = \sigma^2 \left(1 + \sqrt{5}\frac{\|X-X'\|}{\ell} + \frac{5}{3}\frac{\|X-X'\|^2}{\ell^2}\right) \exp\left(-\sqrt{5}\frac{\|X-X'\|}{\ell}\right)$ |
| Linear (or dot product) | No | Once differentiable | $k(t, t') = \sigma^2(t - \gamma) \cdot (t' - \gamma)$ |
| Sampling covariance kernel | Yes | Not applicable | $k(X, X') = Cov(m(X), m(X'))^{\#}$ |
| Polynomial | No | $\lambda$ times differentiable | $k(t, t') = \sigma^2(\gamma + t \cdot t')^{\lambda}$ |
| Constant | No | Not applicable | $k(X, X') = \sigma^2$ |
| Temporal white noise | No | Not applicable | $k(t, t') = \sigma^2\delta(t, t')$ |
| Spatial white noise | Yes | Not applicable | $k(x, x') = \sigma^2\delta(x, x')$ |

[*] All GP kernels can calculate temporal covariance, except the spatial white noise kernel.

† $X$ represents spatio-temporal location, incorporating both the age and spatial coordinates of the data; $t$ denotes the age of the sample; and $x$ indicates the spatial coordinates. $\sigma^2$ = variance; $\ell$ = a positive characteristic length-scale parameter; $\alpha$ = a scale mixture parameter, when $\alpha \to \infty$, the rational quadratic kernel is equivalent to the radial basis function kernel; $\gamma$ = offset or shift parameter, adjusting the baseline level of the kernel's output; $p$ = periodicity parameter for the Periodic kernel, defining the cycle length of repeating patterns; $\lambda$ represents the degree of the polynomial, an integer determining the complexity of the model for the polynomial kernel.

[#] $Cov(m(X), m(X'))$ indicates the sampling covariance between outputs at different spatio-temporal points, derived from deterministic models under varying physical parameter assumptions. Here, $m(X)$ denotes the output at a specific location and time from a suite of physical models assuming different parameters.

supports combining different kernels, either additively, multiplicatively, or both. Additive combinations capture independent contributions from distinct processes, such as long-term trends or periodic variations, treating them as separate effects. In contrast, multiplicative combinations create interactions between processes, resulting in more structured patterns. For example, multiplying a periodic kernel with a linear kernel produces a periodic variation with an amplitude that increases or decreases linearly over time, effectively modeling phenomena where seasonal patterns intensify or diminish progressively (Görtler et al., 2019).

Designed for spatio-temporal data analysis, all GP kernels in PaleoSTeHM support temporal data (represented as a 1-dimensional vector) and most of kernels support spatial data (represented as 2-dimensional matrix including latitude and longitude; see Table 2). Temporal kernel correlations are calculated using the 1-dimensional Euclidean distance between time points, while spatial kernel correlations are derived from the 1-dimensional geographical radial distance, calculated based on the spherical distance between pairs of longitude and latitude under the assumption of a purely spherical Earth geometry. Users can choose to build a temporal or spatial kernel by switching a parameter in each kernel function.

## 3.5 Analysis Choices module

To accommodate diverse computational resources and varying requirements for the trade-off between modeling robustness and computational demands, the Analysis Choices module offers multiple methods for Bayesian inference of model parameters as defined in the Modeling Choices module (Figure 2). This flexibility ensures users can optimize their analyses based on available technology and specific modeling needs. Unlike deterministic methods (e.g., least-squares), which have been extensively implemented in other studies (e.g., Crichton et al., 2023; Lin et al., 2021), PaleoSTeHM focuses on developing Bayesian probabilistic approaches that more effectively manage the inherent uncertainties associated with paleo data.

### 3.5.1 Fully Bayesian analysis

A fully Bayesian analysis requires assigning prior probability distributions to all model parameters, allowing them to take on a range of values, potentially with different probabilities. These priors can either incorporate informative prior knowledge or remain uninformative and vague. Since the posterior distribution is shaped by both the priors and the likelihood of the observed data, it often becomes complex and analytically intractable. *Markov Chain Monte Carlo (MCMC)* methods are crucial in this case as they enable the efficient exploration and approximation of the posterior distribution. PaleoSTeHM supports two advanced MCMC samplers: Hamiltonian Monte Carlo (HMC; Neal et al., 2011) and the No-U-Turn sampler (NUTS; Hoffman et al., 2014), which provide more efficient sampling performance than traditional Metropolis-Hastings MCMC (Hastings, 1970).

HMC significantly improves sampling efficiency over traditional Metropolis-Hastings MCMC by leveraging gradients of the probability distribution to guide the sampling process, which involves generating random samples from the underlying latent probability distribution. This method reduces autocorrelation (the correlation between successive samples in a Markov Chain, indicating how dependent the current sample is on previous ones), thereby increasing the effective sample size (the number of independent samples, accounting for autocorrelation) per iteration (a single step in the sampling process where the algorithm generates a new sample) and enabling faster convergence. Building on HMC, NUTS further enhances efficiency by automatically adapting the path length (the distance traversed in parameter space during a single Hamiltonian trajectory) and managing the step size (the distance traveled in parameter space at each leapfrog step during Hamiltonian dynamics, Bingham et al., 2019). NUTS eliminates the need for manual tuning of these parameters, facilitating more effective exploration of complex, high-dimensional posterior distributions commonly encountered in Bayesian analysis.

Compared to other analysis choices such as empirical Bayesian models or variational Bayesian models (Table 1), a fully Bayesian model offers a more comprehensive estimation of the uncertainties associated with model parameters (Piecuch et al., 2017). It also offers a direct framework for sample age measurement uncertainty in an *errors-in-variable (EIV)* manner, Dey et al., 2000). However, the nature of MCMC-based samplers means they are computationally more demanding. Particularly within the EIV framework, where the number of sampling parameters increases linearly with data size, this leads to a polynomial increase in the computational power required (Belloni and Chernozhukov, 2009), which can be significant and unaffordable when dealing with large datasets or complex models.

### 3.5.2 Empirical Bayesian analysis

Unlike Fully Bayesian analysis, which requires full probability distributions for prior and posterior, Empirical Bayesian analysis offers a practical alternative. This approach approximates a fully Bayesian treatment where parameters at the highest level of the hierarchy are fixed at their most likely values rather than being integrated out. This optimization is typically achieved using the maximum likelihood estimate, leading to a posterior distribution that is conditional on the data and these optimized parameters:

$$p(f|y,\hat{\theta}_s,\hat{\theta}_d) \propto p(y|f,\hat{\theta}_d)p(f|\hat{\theta}_s) \tag{12}$$

here, the posterior probability of the latent processes $f$ is inferred, assuming that the hyperparameters at the data and process levels ($\hat{\theta}_d$ and $\hat{\theta}_s$) are known and fixed. While the existing code base allows for explicit bounds to be set on hyperparameters for the maximum likelihood estimate (e.g., Ashe et al., 2019; Kopp et al., 2016), it does not provide for an explicit prior distribution for the parameters. In other words, it only support a uniformly distributed prior information, limiting the ability to incorporate informative prior knowledge. By leveraging Pyro's variational inference capabilities (details in section 3.5.3), PaleoSTeHM enables users not only to optimize hyperparameters using their maximum likelihood estimate but also to define many commonly-used distributions for each prior model parameter explicitly. This allows optimization to be conducted in a maximum *a posteriori* probability estimation manner, assuming the variational distribution is a Dirac delta function. In PaleoSTeHM, by default, the optimization is achieved using Adam, a stochastic optimizer (Kingma and Ba, 2014). While empirical Bayesian analysis generally requires fewer computational resources than fully Bayesian methods, it is important to note that, assuming hyperparameters at the data and process levels are known and fixed may lead to substantial underestimation in the inference uncertainty (Piecuch et al., 2017).

### 3.5.3 Variational Bayesian analysis

Considering the computational expense required to perform MCMC in fully Bayesian analysis and the limitations of Empirical Bayesian methods that fail to account for the uncertainty of hyperparameters, PaleoSTeHM also supports variational Bayesian analysis, which emerges as an efficient intermediary. Rather than directly sampling from the posterior distribution through MCMC, variational Bayesian methods aim to approximate the true posterior probability distribution ($p(f,\theta_s,\theta_d|y)$) with a simpler, parametric probability distribution ($q(f|\phi)$). Thus, Bayesian inference is transformed from a sampling chal-

lenge into an optimization problem—known as variational inference—requiring significantly fewer computational resources while facilitating uncertainty estimation (Wingate and Weber, 2013).

In PaleoSTeHM, variational Bayesian analysis is achieved by optimizing the variational parameters $\phi$ to minimize the Kullback-Leibler (KL) divergence, a metric to effectively measure the difference between two distributions:

$$\phi = \arg\min_{\phi} \text{KL}\left[q(f, \theta_s, \theta_d | \phi) || p(f, \theta_s, \theta_d | y)\right] \tag{13}$$

For more details above KL divergence, readers can refer to Blei et al. (2017). Adam facilitates this minimization and the variational distribution for PaleoSTeHM is a normal distribution by default. In contrast to MCMC-based fully Bayesian analysis, which often requires computational power that increases polynomially with the number of data points, the optimization-driven approach of variational Bayesian analysis generally scales linearly (Ko et al., 2024; Hoffman and Blei, 2015). Consequently, variational methods can handle larger datasets more effectively, making them suitable for large-scale problems prohibitively for full Bayesian analysis.

### 3.5.4 Incorporation of temporal uncertainty

PaleoSTeHM provides two methods to incorporate temporal uncertainty into final estimations. The first method uses EIV framework (Cahill et al., 2015; Dey et al., 2000), which directly incorporates temporal uncertainty through MCMC sampling of the distribution. The second approach adopts the *noisy-input* framework (McHutchon and Rasmussen, 2011), which applies a first-order Taylor series approximation—a linear expansion around each input point—to account for errors in the independent variable, time, thereby translating these into equivalent errors in the dependent variable:

$$f(x_i, t_i) \approx f(x_i, \hat{t}_i) + \epsilon_i^t \frac{\partial f(x_i, \hat{t}_i)}{\partial t} \tag{14}$$

here $\hat{t}_i$ and $\epsilon_i^t$ are the same as in equation 6, standing for mean observational age and age uncertainty, respectively. The integration of temporal uncertainty within PaleoSTeHM is executed alongside each process level model (Figure 2). All process-level models are implemented using an EIV framework, while for the GP models, both EIV and noisy-input frameworks are available (Figure 2).

### 3.6 Model Validation

After implementing and optimizing a hierarchical model in PaleoSTeHM, it is essential to perform a model validation step to further ensure the robustness and reliability of the trained model. This process involves evaluating how well the model fits the observed data, assessing its predictive accuracy, and diagnosing potential issues such as convergence problems. PaleoSTeHM includes a range of techniques for model validation, such as residual analysis, posterior predictive checks, MCMC convergence diagnostics (e.g., effective sample size and Gelman-Rubin statistic, Gelman and Rubin, 1992), visual inspections (e.g., optimization trace plots, true versus predicted plot), simulation validation and cross-validation methods. These tools allow users to critically examine the model's assumptions, quantify uncertainties, and compare competing models to select the most appropriate one for their specific paleo-environmental application.

To complement these validation techniques, we demonstrate their application in various case studies presented in section 4. Each case study incorporates specific model validation methods tailored to the modeling and analysis choices used. For example, prior and posterior predictive checks are employed to evaluate the performance of optimized models (section 4.1.1); residual plots, weighted mean square error (wMSE) and cross-validation are used to assess the performance of different process-level models (sections 4.1.2 and 4.2); and effective sample size and the Gelman-Rubin statistic ensure good model convergence for MCMC-based analyses (section 4.1.3). Detailed implementations and usage of these validation methods for various Paleo-STeHM experiments, including those mentioned above but not covered in detail in the following sections, are available in the PaleoSTeHM tutorials (see code and data availability).

## 4 Case studies

This section presents illustrative case studies using a tutorial format to demonstrate PaleoSTeHM's usability. All codes and data are accessible and actively managed on the PaleoSTeHM GitHub page (https://github.com/radical-collaboration/PaleoSTeHM). The case studies include:

1. Reconstruction of temporal sea-level changes using coral reef data from the Great Barrier Reef with different data level models (section 4.1.1).

2. Reconstruction of temporal sea-level changes using salt marsh data from New Jersey with different process-level models (section 4.1.2).

3. Reconstruction of temporal sea-level changes using salt marsh data from North Carolina with different Bayesian inference methods (section 4.1.3).

4. Reconstruction of spatio-temporal sea-level changes using various geological proxies from the US Atlantic coast with different process-level models (section 4.2).

Although these examples focus on modeling paleo sea-level, additional tutorials are available for analyzing other paleoenvironmental data, such as ocean temperature anomalies and concentration of carbon dioxide.

The prior and posterior distributions and analysis choice for each model are provided in Table A1. It should be noted this section only briefly describes the modeling results; for a more systematic analysis of paleo-environmental modeling results based on different statistical techniques, the user can refer to Ashe et al. (2019), PAGES2k Consortium (2019) and Tingley et al. (2012).

### 4.1 Time series analysis

#### 4.1.1 Data level modeling

In this section, we examine the impact of the data-level model on inference results. Although numerous paleo-environmental applications commonly assume that proxy reconstruction uncertainties are normally distributed (Ashe et al., 2019; Khan et al.,

2019; Tingley et al., 2012), certain types of proxies may exhibit different forms of uncertainty. For instance, coral reef sea-level proxies indicate past sea-level changes through a quantifiable relationship between the coral's living-habitat depth and the concurrent sea level (Hibbert et al., 2016). Representations of coral living-habitat depth uncertainties are often modeled using either a normal distribution (e.g., Khan et al., 2019) or a uniform distribution (Lin et al., 2021). To illustrate such data-level impact on inference results, we apply a temporally linear model within an EIV framework to coral reef data from the Great Barrier Reef (Yokoyama et al., 2018) using two alternative data-level models. The first can be expressed as:

$$\epsilon_1^y \sim U(\tau_l - \omega_1, \tau_u + \omega_1) \tag{15}$$

where $U$ indicates a uniform distribution between lower and upper ranges defined by specific coral species ($\tau_l$ and $\tau_u$) and an additional noise, defined by hyperparameter $\omega_1$, which follows a prior distribution of:

$$\omega_1 \sim U(10^{-4}, 10) \tag{16}$$

The second data level model can be represented as:

$$\epsilon_2^y \sim N(\mu_2, \sigma_2^2 + \omega_2^2) \tag{17}$$

where $N$ indicates a normal distribution with mean $\mu_2$ and a standard deviation $\sigma_2$, both of which are determined by specific coral species and $\omega_2$ is an additional noise hyperparameter, following the same prior distribution as $\omega_1$. The same prior distributions for each parameter are used for both data-level models, which are represented as non-informative uniform distributions. The characteristics of these non-informative priors are evident in the Prior Predictive Check (Figure A1), which reveals a wide and flat spread of predictions, reflecting the absence of observational influence at this stage.

For both models, the posterior distribution is determined by 11,000 posterior samples drawn from a NUTS sampler, with the first 1,000 samples discarded as burn-in steps. The Posterior Predictive Checks (Figure A1) illustrate that the posterior predictions for both models align closely with the observed data, suggesting successful model convergence. It can be seen in Figure 3 that, although the inference results from different data level models are overall similar, there are still some noticeable differences in the inferred sea-level change trend and rate. Uniform and normal likelihoods yield average sea-level rates of 5.91 mm/yr (4.45-7.38 mm/yr; 90% credible interval, CI) and 6.29 mm/yr (4.81-7.73 mm/yr), respectively. These likelihood assumptions also produce considerably different additional noise parameter distributions. Therefore, users should select an appropriate data-level model to better represent the specific characteristics of different paleoenvironmental data.

### 4.1.2 Process level modeling

To demonstrate the impact of different process level models on inferring paleo sea-level time series, we will use the same data level model together with process level models introduced in section 3.3.2, employing non-informative priors (Table A1). The sea-level data used here is a near continuous core record from single cores of salt-marsh sediment from Leeds Point (New Jersey) covering the Common Era (Kemp et al., 2013). For this database, a normal likelihood data level model is adopted with sea-level reconstruction uncertainties provided by the original study. Here we will test three process level models: (a)

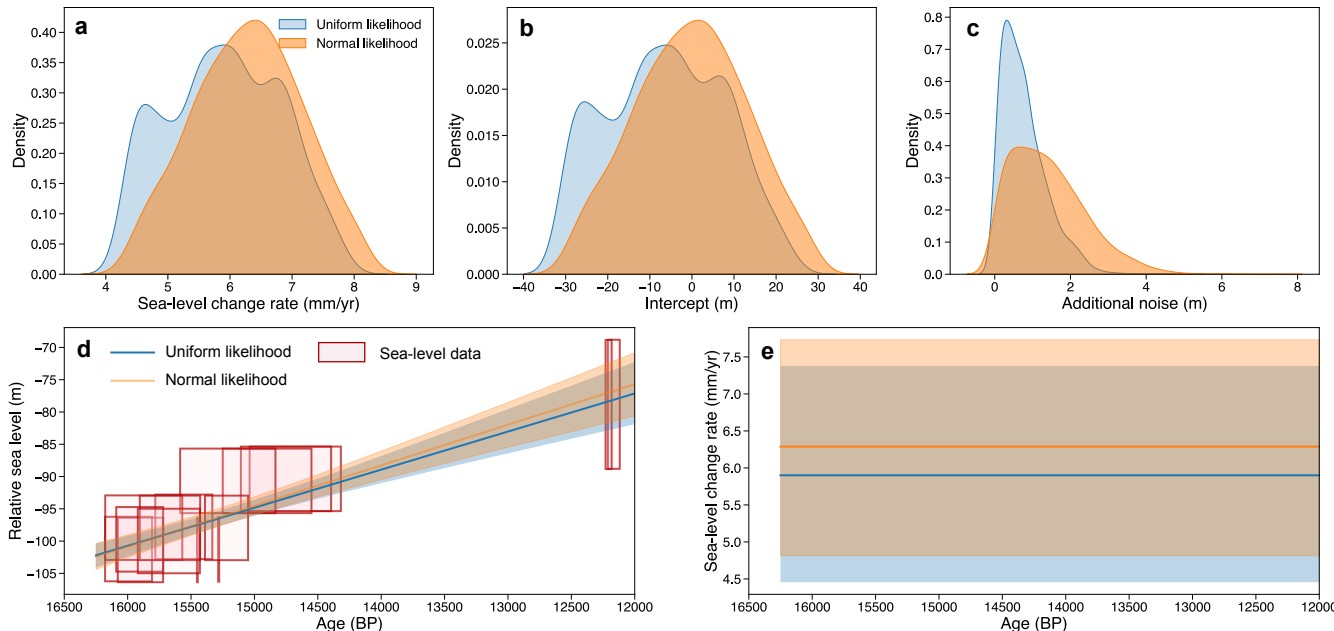

**Figure 3.** Data level models impact on temporal sea-level change inference at the Great Barrier Reef. Posterior probability density functions of sea-level change rate (a), intercept (b) and standard deviation of additional noise (c), assuming either a uniform likelihood (blue) or a normal likelihood (orange). Inferred mean sea-level trends (d) and rates (e) along with a 90% credible interval, where sea-level data are represented by red boxes with horizontal range indicating $\pm 2\sigma$ age uncertainty and vertical range indicating the reconstructed maximum and minimum sea-level range determined by coral species (i.e., $\tau_l$ and $\tau_u$ in equation 15). A negative RSL value indicates that the local RSL in the Great Barrier Reef was lower than present-day levels, reflecting the significant amount of water stored in continental ice sheets. CI = credible interval, BP = before present.

temporally linear model; (b) change-point model (assuming 3 change points; following Ashe et al., 2019); (c) Gaussian Process
model with an radial basis function (RBF) kernel. Posterior distributions for models *a* and *b* were obtained using a variational Bayesian approach, while model *c* employed an empirical Bayesian method.

Figure 4 shows estimated RSL trends and rates of RSL change for each process model. The resulting trends and fit to the data (quantified using wMSE) differ significantly due to the fundamentally different model formulations. The temporally linear model can only estimate an averaged trend and rate of sea-level change and will never predict an accelerated RSL change.
Consequently, it exhibits the highest wMSE (2.53) with a systematic errors with strong temporal correlations displayed in residual plots (Figure A2), both of which reflect a poor fit to the observations.

Comparatively, the change-point model is able to capture a noticeable change in RSL rate from 1.49 mm/yr (1.26-1.70 mm/yr) between -500 CE and 1839 CE (1824-1852 CE) to 3.91 mm/yr (3.72-4.10 mm/yr) after 1839 CE. This added flexibility substantially improves the model's fit to the data, achieving a wMSE of 0.38 and producing a less structured error distribution

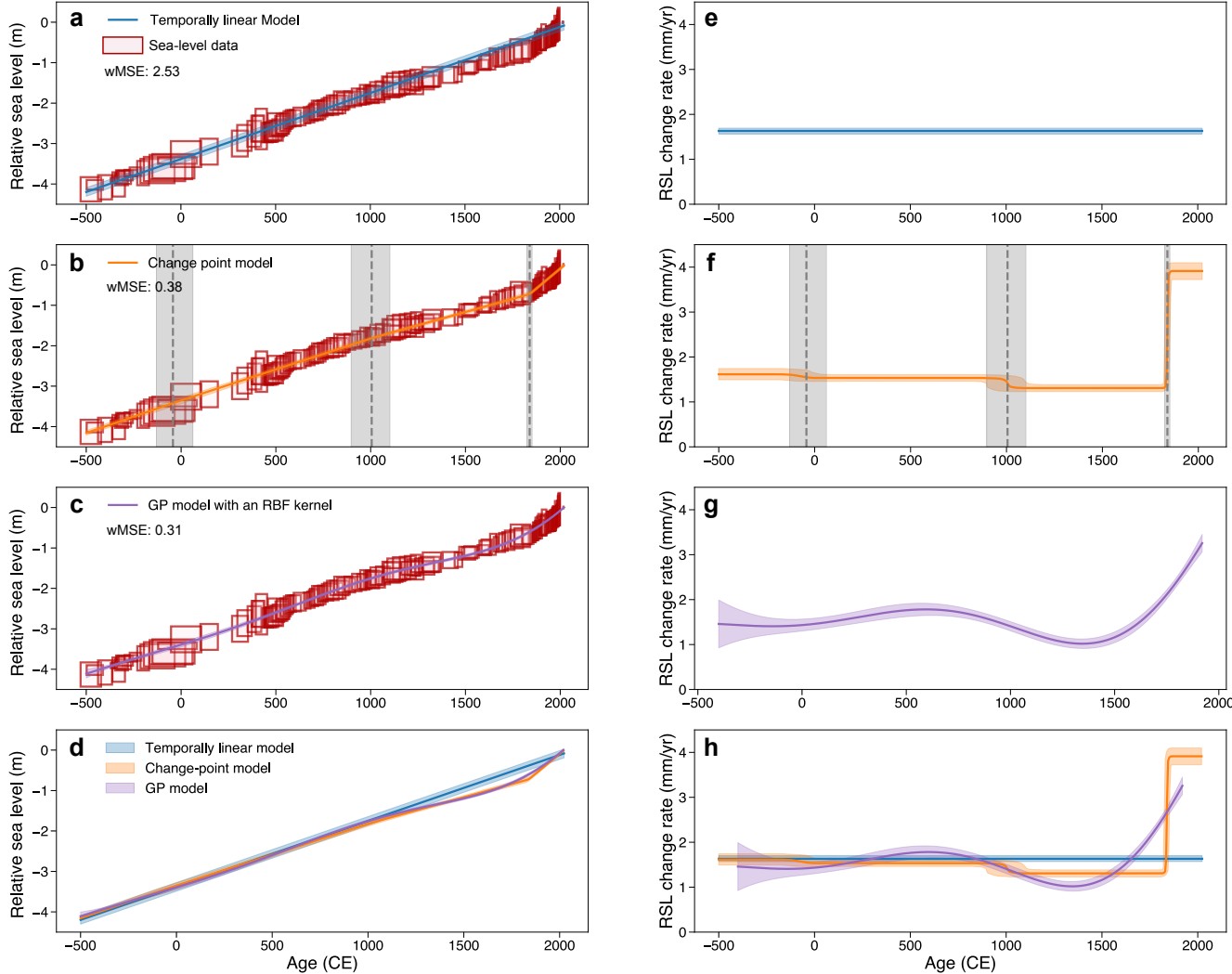

**Figure 4.** Process level models impact on temporal sea-level change inference at New Jersey. Common Era sea-level comparison of linear model (a, e), change-point model (assuming 3 change points; b, f) and Gaussian Process model with an RBF kernel (c, g), where input data are continuous cores. Output includes estimates of RSL (a-d) and rates of RSL change (e-h), which are each shown with mean and 90% credible intervals. Paleo sea-level data here are modeled using a normal likelihood. The horizontal and vertical range of red boxes indicates $\pm 2\sigma$ age and relative sea-level reconstruction uncertainties, respectively. Sea-level data here were reconstructed using near-continuous record from single cores of salt-marsh sediment from Leeds Point (New Jersey, Kemp et al., 2013). wMSE = weighted mean square error; CE = Common Era.

(Figure A2). Such flexibility makes the change-point model particularly suitable for identifying the time of emergence in various environmental change contexts (e.g., Walker et al., 2022; Caesar et al., 2021; Lyu et al., 2014).

As a non-parametric approach, the GP model produces continuous distributions of RSL change rates over time, allowing for the estimation of multiple inflection points (Walker et al., 2022). This flexibility results in the lowest wMSE (0.31), alongside minimal temporal structure in the residuals (Figure A2), indicating the best overall fit to the observations. However, the infinite differentiability of the RBF kernel can lead to overly smooth predictions in time series analysis, potentially oversmoothing sharp changes that are critical in many environmental contexts, such as abrupt sea-level rise (Lin et al., 2021), ocean circulation slowdowns (Caesar et al., 2018), and extreme events like heavy rainfall (Stein, 2012). Alternative kernels (e.g., Matérn kernels) can provide alternative levels of differentiability.

### 4.1.3 Analysis choices

Using similar near *continuous core* data from Sand Point, North Carolina (Kemp et al., 2011), we illustrate the effects of analysis choices on RSL inference. Here, we only use a subset of the original data to better demonstrate the difference between various analysis choices. The adopted data and process level model employ a normal likelihood with a GP model using an RBF kernel (Table A1). The hyperparameters will be sampled using empirical, fully Bayesian and variational Bayesian methods. For the fully Bayesian method, the posterior distribution is determined by 5,500 posterior samples drawn from a NUTS sampler, with the first 200 samples discarded as burn-in steps. For the empirical and variational Bayesian methods, the hyperparameters were optimized using the Adam optimizer over 1000 iterations (Kingma and Ba, 2014). The run times of each implementation are reported on a 2023 MacBook Pro with an Apple M2 Pro chip.

For MCMC-based fully Bayesian analysis, PaleoSTeHM employs the Gelman-Rubin statistic (Gelman and Rubin, 1992) to verify that the Markov chains have converged to a stationary phase, indicating good convergence. Additionally, the effective sample size is used to assess the amount of information retained, accounting for the correlation in the sequence (Bürkner, 2017). Typically, a Gelman-Rubin statistic of less than 1.1 and an effective sample size greater than 1000 suggest reliable sampling of the posterior distribution. In this case, the analysis meets these criteria with a Gelman-Rubin statistic of 1.0 and an effective sample size exceeding 3000. For empirical and variational Bayesian methods, validation is typically conducted through the inspection of optimization trace plots (plots showing how optimization target function improve with each iteration), where successful optimization is characterized by a steadily decreasing loss function and parameters convergence over iterations. These conditions are also satisfied in this analysis, as illustrated in the corresponding optimization trace plots (Figure A3 and A4).

Figure 5 compares posterior distributions of RSL trend and rate of change and the computational time for each analysis choice. The empirical Bayesian method requires the least computational power, only providing a point estimate of hyperparameters without accounting for their underlying uncertainty. Although more computationally demanding, the fully Bayesian method captures the hyperparameter uncertainties effectively. As an intermediary, variational Bayesian method requires slightly more computational time compared to empirical method but can derive a variational posterior distribution that is largely similar to that obtained by the fully Bayesian method through MCMC sampling. In contrast, the point estimate obtained through the empirical Bayesian method falls at the third percentile of the posterior hyperparameter distributions derived from the fully Bayesian method, highlighting a significant bias introduced by the overly simplistic approach.

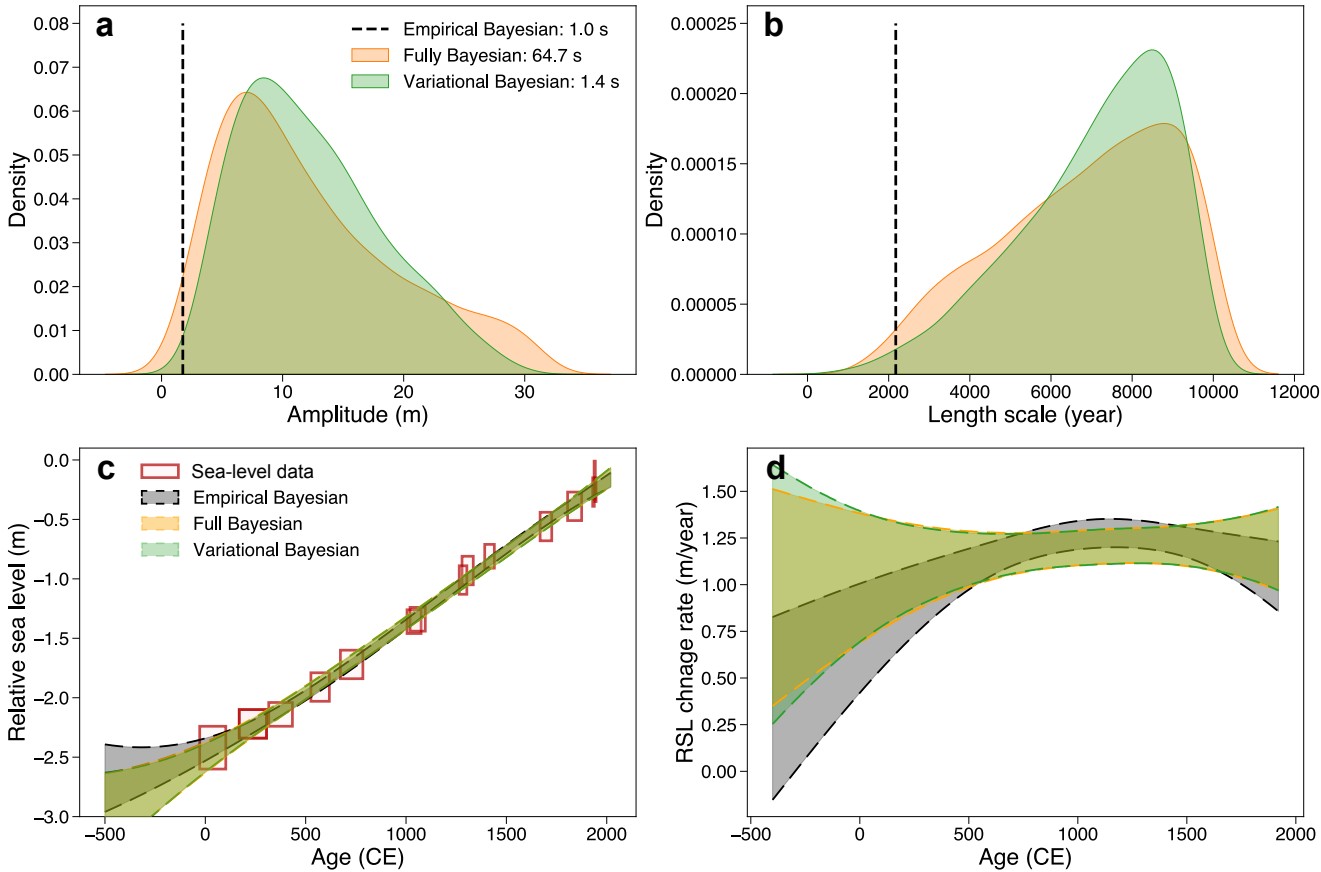

**Figure 5.** Analysis choices impact on temporal sea-level change inference at North Carolina. (a-b) GP model hyperparameters optimization results along with the required computational time in second (based on a 2023 MacBook Pro with an Apple M2 Pro chip). (c-d) Common Era sea-level comparison between three analysis choices, the results indicate 90% credible interval of RSL change trend (c) and rate (d). Paleo sea-level data here are modeled using a normal likelihood. The horizontal and vertical range of red boxes indicates $\pm 2\sigma$ age and relative sea-level reconstruction uncertainties, respectively. Sea-level data here were reconstructed using near-continuous record from single cores of salt-marsh sediment from Sand Point (North Carolina, Kemp et al., 2011)

Because of the near continuous sea-level data with smoothly rising sea-level trend in North Carolina, the inference results from these three methods are similar. However, given that geological sea-level data are often sparsely distributed across both spatial and temporal domains and may subject to abrupt change in rate, neglecting the underlying uncertainty of hyperparameters by empirical Bayesian method can result in a significant underestimation of the final inference uncertainty compared with fully Bayesian method.

## 4.2 Spatio-temporal analysis

Spatio-temporal analysis presents a common challenge in paleo-environmental studies, such as reconstructing continuous spatio-temporal signals from sparse and noisy data. To address this, PaleoSTeHM provides a range of approaches, spanning purely statistical to purely physical methods. Here, we present an illustrative example to recover the spatio-temporal RSL pattern and its associated uncertainty. This analysis utilizes a sea-level database containing 1,043 proxy records spanning from 11 ka to the present, compiled by Ashe et al. (2019) from previous studies (Kemp et al., 2017a, b, 2015, 2014, 2013; Khan et al., 2017; Engelhart and Horton, 2012). The database includes sea-level proxies such as salt marsh, mangrove, beach rock, and coral. All records were used to train the model except for 51 sea-level data points from New York (Engelhart and Horton, 2012), which were reserved for cross-validation—a technique used to evaluate model performance on unseen data (shown as a gray dot in Figure 6m-p).

We demonstrate four process level models that have been used in previous studies: (i) a GP model with a zero mean function and multiple isotropic kernels (Ashe et al., 2019); (ii) a GP model with the mean function determined by a GIA model and multiple isotropic kernels (Walker et al., 2021; Kopp et al., 2016); (iii) a GP model with a zero mean function and a sampling covariance kernel determined by a GIA model ensemble (Kopp et al., 2009); and (iv) a purely GIA model ensemble (Lin et al., 2023a). All models assume a data-level model with a normal likelihood determined by RSL reconstruction uncertainty and an additional white noise term. For this analysis, we implement a noisy-input framework to address temporal uncertainty and use the empirical Bayesian method to optimize hyperparameters for models *i*, *ii* and *iii*, while model *iv* is optimized through the variational Bayesian method (Table A1).

For model *i*, we follow the kernel structure as in Ashe et al. (2019), which can be expressed as:

$$f(X) \sim GP(0, K_1(X, X')) \tag{18}$$

$$K_1(X, X') = g(t) + r(x, t) + l(x, t) \tag{19}$$

where $g(t)$ represents a spatially-uniform covariance function, while $r(x, t)$ and $l(x, t)$ are regional and local varying isotropic covariance functions, respectively. These are characterized by a 3/2 Matérn temporal kernel (Table 2) for $g(t)$ and a product of a 3/2 Matérn temporal kernel and a 1/2 Matérn spatial kernel 2) for $r(x, t)$ and $l(x, t)$, which are distinguished by their prior distributions of hyperparameters.

Similarly, model *ii* can be written as:

$$f(X) \sim GP(\text{GIA}(X), K_2(X, X')) \tag{20}$$

$$K_2(X, X') = r(x, t) + l(x, t) \tag{21}$$

here, the mean RSL expectation is determined by RSL prediction from ICE_7G ice model with VM5a Earth model (Roy and Peltier, 2018; Peltier et al., 2015), and $r(x,t)$ and $l(x,t)$ are the same as in equation 19. We do not include $g(t)$ kernel here as the RSL prediction derived from the ICE_7G model—embedded within the GP mean function and rigorously calibrated against comprehensive RSL and geodetic datasets across North America Roy and Peltier (2018); Peltier et al. (2015)—is assumed to adequately capture all spatially uniform signals.

Model *iii* can be denoted as:

$$f(X) \sim GP(0, K_3(X, X'))$$ (22)

$$K_3(X, X') = Cov(m(X), m(X')) \cdot exp(-|t - t'|/\tau^2)$$ (23)

here, $Cov(\cdot)$ here indicates a sampling covariance function through a physical model ensemble $m$. In this context, the covariance between data points is not directly determined by their spatio-temporal proximity but instead depends on the variance within the physical model ensemble, conditioned on various combinations of physical parameters. For this *iv*, $m$ includes an ensemble of forward GIA models: (a) ICE_7G ice model with VM5a Earth model; (b) PaleoMIST ice model (Gowan et al., 2021) with 71 km lithosphere and 0.3 and $70 \times 10^{21}$ Pa s upper and lower mantle viscosity; and (c) ANU ice model (Lambeck et al., 2014) with 71 km lithosphere and 1 and $10 \times 10^{21}$ Pa s upper and lower mantle viscosity. To expand the variability of physical model predictions, we create six synthetic GIA model outputs by enlarging or shrinking these three GIA model outputs by 1.5. Therefore, this physical model ensemble consists of predictions from nine models. More details about the physics-based GIA model used here can be found in Lin et al. (2023b). To stabilize the estimate and reduce variability related to finite sample size, we applied a temporal Gaussian taper function to this kernel, controlled by a parameter $\tau$. Following Hay et al. (2015); Kopp et al. (2009), we set $\tau$ to 3000 years.

And lastly, model *iv* can be written as a weighted mean of different physical models:

$$f(X) = \sum_{n=1}^{N} \nu_i \text{GIA}_i(X)$$ (24)

$$\nu \sim \text{Dirichlet}(\alpha_d)$$ (25)

In this model, $\nu$ represents the relative weights associated with each GIA model. These probabilities follow a Dirichlet distribution (or multivariate beta distribution) characterized by a concentration parameter $\alpha_d$. A value greater than 1 for $\alpha_d$ indicates a preference for a more evenly distributed probability across all models. In contrast, a value less than 1 indicates a preference for more concentrated probabilities on fewer models (Lin et al., 2023b). For this experiment, we set $\alpha_d$ according to each GIA model prediction fit to RSL observation (using weighted root mean square as a metric, see Table A1).

A comparison of RSL inference results between different spatio-temporal process level models is provided in Figure 6. At the purely statistical end of the process model spectrum, model *i* correlates RSL from various locations and times based

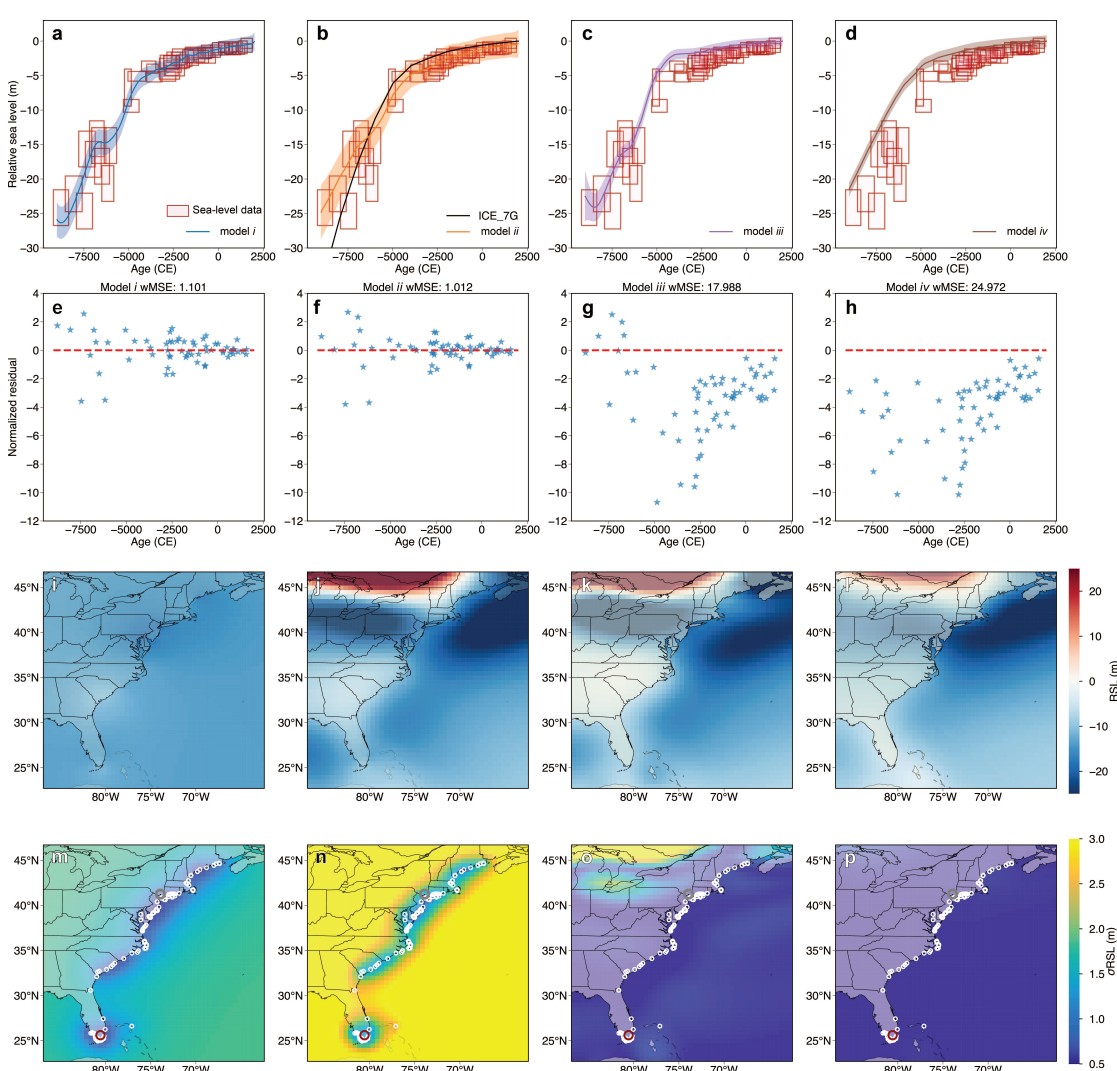

**Figure 6.** The impact of process level modeling choices for spatio-temporal sea-level change inference along the US Atlantic coast. Each column represents a different process level model. (a-d) Mean and 90% credible or confidence interval for RSL predictions at Florida (indicated by a red dot in m-p). The horizontal and vertical range of red boxes indicates $\pm 2\sigma$ age and relative sea-level reconstruction uncertainties, data from Khan et al. (2017). (b) For Model *ii*, the GP mean function, determined by the ICE_7G model (Roy and Peltier, 2018; Peltier et al., 2015), is depicted with a black line. (e-h) Normalized residuals, which represent the difference between observed and predicted values normalized by observational uncertainty, for Florida RSL predictions generated by each process-level modeling choice. Red dashed lines represent 0 error. The weighted mean square error for each model is given in title. (i-l) Mean RSL prediction for the year -5,500 CE. (m-p) Standard deviation of the RSL prediction for the year -5,500 CE, where white dots indicate locations where each sea-level data collected and gray dot represent location of New York where 51 data points were held for testing purpose.

solely on their spatio-temporal proximity, a property derived from the adopted isotropic kernels. According to model *i*, the RSL change along the US Atlantic coast during the Holocene was dominated by a spatially uniform signal (produced by the spatially-uniform kernel, $g(t)$, in equation 19; Table A1), which contributed to more than 25 m of RSL rise. In contrast, $r(x,t)$ and $l(x,t)$ only produce up to 5 m of spatially variable RSL signal, resulting in virtually no spatial pattern in the mean RSL prediction of this model. In the temporal domain, multiple studies have demonstrated that GP models like model *i* can accurately recover multi-millennial sea-level variation trends at locations with abundant sea-level observations, such as Florida, as shown in Figure 6a (Tang et al., 2023; Ashe et al., 2019; Cahill et al., 2015). This is further supported by the residual plot (Figure 6e), which exhibits a low wMSE (1.1) and minimal temporal structure in the residuals, indicating a good agreement with the observed data.

However, spatial inferences based on isotropic and stationary kernels of model *i* are often considered overly simplistic (Stein, 2005a), partly due to the sparse nature of geological data and the complexity of environmental change mechanisms. As see in Figure 6m, geological sea-level data are mostly collected across paleo-coastal areas. Therefore, RSL inferences from model *i* are only representative of coastal areas (as opposed to terrestrial or marine areas) and cannot adequately reflect the physical knowledge of paleo sea-level change (e.g., the RSL uncertainty caused by the existence of the Laurentide Ice Sheet).

Model *ii* uses a deterministic GIA model (ICE_7G ice model with VM5a Earth model, Roy and Peltier, 2018; Peltier et al., 2015) as the GP mean function. By harnessing a physics-based model, model *ii* captures intricate spatial sea-level variation patterns due to the GIA-induced GRD effects (Figure 4). In this setup, the covariance functions describe residuals between the GIA model and RSL observations (mostly captured by $r(x,t)$ in equation 21). Similar to model *i*, RSL predictions for Florida by model *ii* closely align with observations, demonstrating a low wMSE (1.0) and an unstructured residual distribution. In the spatial domain, at -5500 CE, model *ii* suggests the GIA model underestimates ∼10 m RSL at New Jersey (Figure A5), which may reflect oversimplified physics (e.g., neglecting 3D solid Earth rheology; Austermann et al., 2013), biased sampling of physical parameters (such as poorly-constrained ice history), or missing physical processes in the GIA model (e.g., sediment isostatic adjustment; Lin et al., 2023a). Because model *ii* assumes no uncertainty in GIA modeling, the uncertainty quantification here also relies solely on the radial distance from RSL data points (Figure 6n).

Model *iii* utilizes a kernel constructed by sampling covariances between various forward GIA models based on alternative ice and Earth models. By incorporating relevant physical processes into GP kernel construction, model *iii* effectively captures anisotropic behaviors, non-stationarities, heterogeneities, and *teleconnections* that are intrinsic to the physical dynamics of RSL change but are challenging to represent with standard classes of covariance functions (Table 2). For instance, the size of the Laurentide Ice Sheet exhibits a positive correlation with RSL around the northern Great Lakes while displaying a negative correlation with RSL in peripheral bulge regions such as New Jersey (Figure 6k).

Model *iii* also has certain limitations, including the computational cost of thoroughly sampling physical model parameters and the presence of structural errors within the physical models, such as oversimplifications or omitted processes. As shown in the residual plot of Figure 6g, there is a significant mismatch between RSL observations and model *iii* predictions, reflected in a high wMSE value (17.99) and pronounced temporal structure in the residuals. This poor fit may stem from biased sampling of ice and Earth models, as well as the reliance on an oversimplified 1-D rheology. Furthermore, the model prioritizes fitting

regions with denser data distribution, such as the mid-northern US Atlantic coast. For instance, model *iii* provides a good fit to unseen RSL observations in New York (Figure 7c,g), but this prioritization comes at the expense of accuracy in regions with sparser data, such as Florida, where substantial misfits are observed. Additionally, the posterior mean and standard deviation generated by this method are less directly interpretable compared to those produced by model *iv*.

Model *iv* represents the purely physical end of the process-level spectrum and is formulated as a weighted linear combination of physical models, with weights determined by data-model misfits (e.g., wMSE and chi-square misfit; Lin et al., 2021; Li et al., 2020; Lambeck et al., 2014). The mean and uncertainty estimated by this method reflect the parametric uncertainty inherent in a given physical model, allowing for direct interpretation of physical parameters, such as deriving posterior distributions of global ice history (Creel et al., 2024). However, this approach is also susceptible to structural errors within the model, similar to those observed in model *iii*. The limited sample size of physical parameters—only nine models were used in this analysis—and the model's tendency to prioritize fitting denser sea-level data in mid-northern locations result in uncertainty estimates that appear underestimated and biased, as illustrated by substantial misfits to observations (Figures 6d and h). Furthermore, the difficulty in directly quantifying certain physical parameters often leads to oversimplified model predictions. For example, the scarcity of direct constraints on ice history (Dalton et al., 2020) reduces the ability of forward GIA models to resolve centennial-scale sea-level variations. This limitation makes these models less effective in capturing centennial-scale variability compared to models *i–iii* (Figure 6).

Due to the dense distribution of sea-level data along the mid-northern US Atlantic coast, all models effectively capture the general trend of RSL variation observed in the withheld data from New York (Figure 7), as evidenced by low wMSE values and minimal temporal structure in the residual plots. While model *ii* achieves the lowest wMSE and model *iv* exhibits slight bias in its residuals, all models demonstrate comparable overall performance in reconstructing sea-level changes. In contrast, significant variation in model performance is observed for RSL predictions at Florida (Figure 6a–h), where models *iii* and *iv* show substantial misfit to RSL observations. This misfit stems from the requirement to preserve the overall consistency of the physical systems constrained by the ensemble of physical models.

It is important to note that high-quality and standardized datasets, such as those available for the mid-northern US Atlantic coast, are rare in many paleoenvironmental fields, such as deep-sea isotopes or ice core records (Shackleton et al., 2021; Lemieux-Dudon et al., 2010). Consequently, users must carefully evaluate factors such as data availability, computational resources, the need for interpretability, and the level of understanding of underlying physical processes when selecting a process level model. Generally, physics-based models offer superior interpretability and better extrapolation capabilities to spatio-temporal locations with minimal data, as they are rooted in well-established theoretical frameworks. However, discovering and validating new physical laws can be time-intensive and often computationally demanding. In contrast, machine learning or statistical approaches provide flexibility and computational efficiency but often face challenges in extrapolating non-linear functions (Xu et al., 2020; Goodfellow et al., 2016). Whereas, they require large volumes of training data and rigorous validation to ensure consistency with physical principles. For a more detailed discussion on the integration of physics-based and machine learning models, readers are referred to Lai et al. (2024).

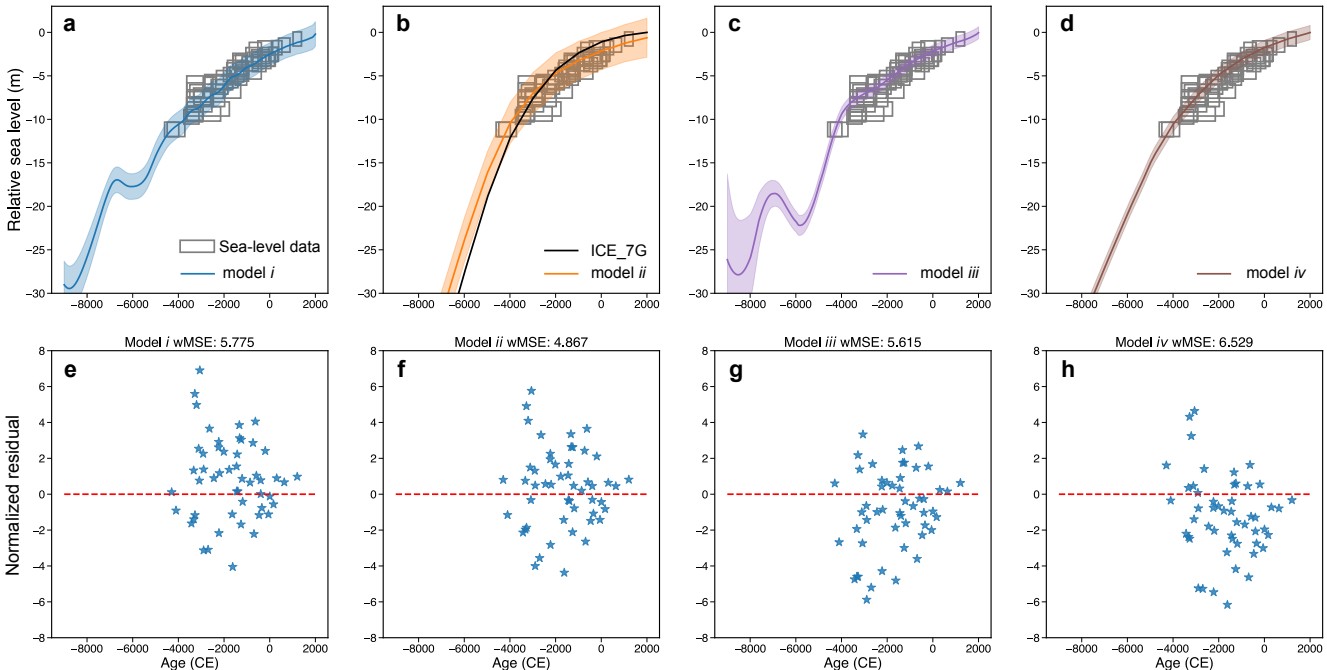

**Figure 7.** Performance of different process-level modeling choices on unseen sea-level data from New York. (a-d) Mean and 90% credible or confidence intervals for RSL predictions at New York (gray dot location shown in Figure 6m-p). The gray boxes represent $\pm 2\sigma$ uncertainties in both age and relative sea-level reconstructions, with data from Engelhart et al. (2009). For Model *ii*, the GP mean function, derived from the ICE_7G model (Roy and Peltier, 2018; Peltier et al., 2015), is shown as a black line (b). (e-h) Normalized residuals, which represent the difference between observed and predicted values normalized by observational uncertainty, for New York RSL predictions generated by each process-level modeling choice. Red dashed lines represent 0 error. The weighted mean square error for each model is given in title.

## 5    Discussion

### 5.1    Generalization for paleo-environmental problems

This paper focuses on demonstrating the functionality of PaleoSTeHM in paleo-sea-level applications. However, the flexibility of hierarchical models, where any statistical model can be interpreted hierarchically, allows the PaleoSTeHM framework to be readily applicable to a wide range of paleo-environmental problems. By using hierarchical models, transparency is enhanced by distinguishing between modeling assumptions and analytical methods, as well as separating process variability from observational noise.

The common characteristics of paleo sea-level datasets, such as sparsity and discreteness, are shared by many other paleo-environmental datasets, including paleo temperature (PAGES2k Consortium, 2017), past ice sheet thickness (Small et al., 2019), and sediment deposition depth (Wang et al., 2018). As a result, the data and process-level models introduced in this paper can be readily generalized to these paleo-environmental fields. For example, Tingley et al. (2012) and Stein (2005b)

proposed that using a GP model is a reasonable approach to describe latent space-time climate processes, such as annual mean surface temperature anomalies and daily wind speed; Lin et al. (2023a) applied a spatial GP model to recover the spatial pattern of Holocene coral reef depth based on a Holocene coral reef deposition depth database across the Great Barrier Reef (Hinestrosa et al., 2022); and Caesar et al. (2018) implemented a change-point model on multiple proxy datasets to detect significant reductions in the strength of the Atlantic Meridional Overturning Circulation.

Beyond the process level models featured in PaleoSTeHM v1.0, various approaches have been employed for paleo-environmental analyses. Common techniques for addressing problems in this field include principal component analysis, equivalent to the empirical orthogonal function method when temporal aspects are considered, autoregressive models and generalized additive models. For instance, Shakun and Carlson (2010) used an empirical orthogonal function approach to detect modes of deglacial temperature variability and Piecuch et al. (2017) adopted a degree-1 autoregressive model to reconstruct sea-level evolution using tide gauge data, Simpson (2018) and Upton et al. (2023) developed a series of generalized additive models to model paleo-ecology and paleo sea-level, respectively. While the reimplementation of these models in PaleoSTeHM is beyond the scope of this paper, doing so would benefit from the framework's multiple analysis options and its capacity for smooth integration with flexible data and parameter-level models.

## 5.2 Future developments

From a scientific perspective, numerous promising directions exist for further development of PaleoSTeHM.

Existing data-level models only support a common class of likelihood. However, in paleo-environmental studies, it is typical for proxy data to be subject to complex likelihoods (Ashe et al., 2022; Hibbert et al., 2016). For instance, organic matter that has been radiocarbon-dated undergoes a calibration procedure to account for the time-evolving atmospheric carbon concentration, which can yield a data chronology characterized by multi-modal distributions that significantly differ from each other. Similarly, it is common for paleo-environmental studies to use multiple types of proxy data with different likelihoods to infer a common signal. Recently, new approaches have been developed to account for non-parametric proxy distributions within a hierarchical modeling framework (e.g., Ashe et al., 2022), which could better characterize the underlying uncertainty but can be computationally expansive.

While PaleoSTeHM allows users to specify any number of change points ($m$ in equation 10) in the model, determining the optimal number of change points can be challenging and may require additional modeling strategies. Recent advancements, such as Bayesian transdimensional models (e.g., Sambridge, 2016; Bodin et al., 2012; Gallagher et al., 2011), provide a flexible framework by treating the number of change points as an unknown parameter, allowing it to be inferred alongside other model parameters. Incorporating such approaches into PaleoSTeHM is a potential avenue for future development to address this complexity in abrupt paleoenvironmental change problems.

The current GP Kernel module incorporates commonly used kernel options that are stationary, isotropic and *space-time separable*. While these assumptions simplify calculations significantly, they may not be suitable for some environmental applications. For example, temperature and dew point variations often exhibit strong non-stationary behavior influenced by diverse geographic and atmospheric conditions (Poppick and Stein, 2014). Additionally, the assumption of stationarity may cause rate

uncertainty estimates to fail in properly reflecting the reduced uncertainty expected during periods with abundant data, as shown in Figure 5d, largely because the model's variance is uniformly applied across the entire temporal domain (Heinonen et al., 2016). Furthermore, temperature anomalies over the last two millennia (Mann et al., 2008) demonstrate strong space-time interactions, which cannot be captured by a space-time separable kernel (Tingley et al., 2012). Developing a scientifically richer class of kernel structures could be an important future advancement for PaleoSTeHM. However, given the fundamental differences across various paleo-environmental problems, generalizing sophisticated kernel structures to multiple fields remains challenging.

Another outstanding issue for GP based process level models is scalability, the standard GP models included in PaleoSTeHM v1.0 cannot scale well to large data sets (>10 thousands data points) due to the computational cost, which increases at a rate of $\mathcal{O}(n^3)$, where $n$ is the number of data points (Hensman et al., 2013). Thus, implementing alternative classes of GP models within PaleoSTeHM to model large data sets, especially when incorporating modern environmental observations, which often consist of millions of data points, is an important next step for PaleoSTeHM to develop in the future. Some potentially efficient methods include sparse GP (Quinonero-Candela and Rasmussen, 2005), stochastic variational GP (Hensman et al., 2013) and exact GP with black-box matrix-matrix inference (Wang et al., 2019).

Building upon machine learning infrastructure, another promising direction for the future development of PaleoSTeHM is integrating spatio-temporal hierarchical modeling with machine learning-based emulators as a process-level model. An emulator indicates a statistical model that mimics the behavior of the physics-based simulator but is computationally cheap to run (Reichstein et al., 2019), which is particularly useful for fast sensitivity analysis, model parameter calibration and derivation of confidence intervals for the estimate. The use of statistical emulators trained by physical models will enable hierarchical models to capture the non-stationary physical systems better and enable better interpretation of the modeling results. For paleo-environment, Holden et al. (2019) presents a GP-based emulator for an atmosphere-ocean general circulation model with intermediate-complexity and Lin et al. (2023b) developed a neural network-based emulator for GIA-induced global sea-level change.

More broadly, PaleoSTeHM has been developed by a small team specialized in modeling paleo sea-level changes over multi-millennial time scales. Moving forward, a critical objective is to expand PaleoSTeHM into a larger-scale paleo-environmental community project, where modules are developed autonomously by diverse research teams. The design of PaleoSTeHM, which allows modules to act as wrappers for independently developed code, is specifically intended to facilitate this collaborative effort.

## 6  Conclusion

Paleo-environmental records provide critical out-of-sample information essential for contextualizing current global changes and testing models used to simulate future environmental scenarios. However, our understanding of past environmental changes is often complicated by the sparse nature of geological records, geochronological uncertainties and the indirect relationships between proxies and ecological variables. Hierarchical modeling offers a conceptually straightforward framework to address

these challenges, though the limited availability of user-friendly software often hinders it. PaleoSTeHM offers a flexible and open-source platform that facilitates the rapid and easy implementation of hierarchical models for paleo-environmental applications. The inclusion of multiple process-level models in PaleoSTeHM allows it to be readily applicable across a broad spectrum of paleo-environmental studies. Additionally, its flexibility allows for customization to meet the specific needs of diverse paleo-environmental problems, such as using different Gaussian Process kernels or substituting alternative process-level models.

*Code and data availability.* The development version of PaleoSTeHM is available under an MIT license in a Git version-controlled repository at https://github.com/radical-collaboration/PaleoSTeHM (last access: 16 January 2025). The latest release is archived on Zenodo with the identifier https://doi.org/10.5281/zenodo.12730141 (Lin et al., 2024). Documentation of PaleoSTeHM is available at https://paleostehm.org/. All codes required to generate results and figures shown in section 4 are available in the repository. Video tutorials are available at https://youtube.com/playlist?list=PLR4-1Y89NM_x3zwnxc5nI2mU3pplGzIa3&si=5VoDvpZAWwLE2by4.

*Author contributions.* YL developed the PaleoSTeHM architecture and modules. REK and SJ conceived the project and REK supervised and administered it. AR guide software engineering of PaleoSTeHM, ELA provided YL guidance in statistical modeling. All authors contributed to code development and the writing and editing of the paper.

*Competing interests.* The authors declare no competing interests.

*Acknowledgements.* We thank Chris Piecuch, Roger Creel, Fiona Hibbert, Jeremy Ely, Sönke Dangendorf and the participants of PaleoSTeHM workshops for their valuable feedback in the development of PaleoSTeHM. Yucheng Lin, Robert E. Kopp, Alexander Reedy and Shantenu Jha are supported by the U.S. National Science Foundation under awards 2002437 and 2148265. The authors acknowledge PALSEA, a working group of the International Union for Quaternary Sciences (INQUA) and Past Global Changes (PAGES), which received support from the Swiss Academy of Sciences and the Chinese Academy of Sciences.

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

# Appendix A:  Additional model information

**Table A1.** Summary of Model Characteristics. The posterior is reported with a mean value with 90% credible interval. GBR = Great Barrier Reef, NJ = New Jersey, NC = North Carolina, EIV = errors in variable, NI = noisy input.

| Task | Analysis Choice | Data Level | Process Level | Prior for Parameters | Posterior |
|---|---|---|---|---|---|
| GBR coral time series | Fully Bayesian; EIV | Uniform likelihood with additional white noise | Temporally linear | $\alpha \sim U(-30, 30)$ m $\beta \sim U(-10, 10)$ mm/yr $\omega \sim U(0.0001, 10)$ m | -6.3 (-28.3, 16.2) 5.9 (4.5, 7.4) 0.8 (0.1, 2.0) |
| GBR coral time series | Fully Bayesian; EIV | Normal likelihood with additional white noise | Temporally linear | $\alpha \sim U(-30, 30)$ m $\beta \sim U(-10, 10)$ mm/yr $\omega \sim U(0.0001, 10)$ m | -0.2 (-22.8, 21.9) 6.3 (4.8, 7.7) 1.4 (0.01, 1.97) |
| NJ salt marsh time series | Variational Bayesian; EIV | Normal likelihood | Temporally linear | $\alpha \sim U(-5, 5)$ m $\beta \sim U(-10, 10)$ mm/yr | -3.38 (-3.47, -3.30) 1.63 (1.57, 1.69) |
| NJ salt marsh time series | Variational Bayesian; EIV | Normal likelihood | Change-point model | $\alpha_1 \sim U(-15, 0)$ m $\beta_1 \sim U(-10, 10)$ mm/yr $\beta_2 \sim U(-10, 10)$ mm/yr $\beta_3 \sim U(-10, 10)$ mm/yr $\beta_4 \sim U(-10, 10)$ mm/yr $\gamma_2 \sim U(-476, 1020)$ CE $\gamma_3 \sim U(23, 1518)$ CE $\gamma_4 \sim U(521, 2017)$ CE | -4.92 (-5.01, -4.84) 1.6 (1.5, 1.8) 1.53 (1.46, 1.60) 1.31 (1.22, 1.39) 3.92 (3.73, 4.10) -42.0 (-139.6, 58.2) 1004.3 (893.1, 1106.0) 1838.0 (1823.6, 1853.2) |
| NJ salt marsh time series | Empirical Bayesian; NI | Normal likelihood | Gaussian Process with one RBF kernel | $\ell \sim U(1, 5000)$ yr $\sigma \sim U(1, 22.4)$ m | 1038 20.79 |
| NC salt marsh time series | Empirical Bayesian; NI | Normal likelihood | Gaussian Process with one RBF kernel | $\ell \sim U(1, 10000)$ yr $\sigma \sim U(1, 100)$ m | 2175 1.75 |
| NC salt marsh time series | Fully Bayesian EIV | Normal likelihood | Gaussian Process with one RBF kernel | $\ell \sim U(1, 10000)$ yr $\sigma \sim U(1, 100)$ m | 6889 (2845, 9766) 12.5 (3.4, 28.1) |
| NC salt marsh time series | Variational Bayesian; NI | Normal likelihood | Gaussian Process with one RBF kernel | $\ell \sim U(1, 10000)$ yr $\sigma \sim U(1, 100)$ m | 7305 (3559, 9509) 10.4 (3.8, 18.4) |

| Task | Analysis Choice | Data Level | Process Level | Parameter Level | Posterior |
|---|---|---|---|---|---|
| US Atlantic spatio-temporal analysis | Empirical Bayesian; NI | Normal likelihood with additional white noise | Gaussian Process with a zero mean function and multiple isotropic kernels | $\omega \sim U(0.01, 10)$ m | 0.02 |
| | | | | $\ell_g \sim U(100, 20000)$ yr | 11567 |
| | | | | $\sigma_g \sim U(0.01, 33.3)$ m | 30.7 |
| | | | | $\sigma_r \sim U(0.2, 10)$ m | 1.7 |
| | | | | $\ell_{r,x} \sim U(319, 1593)$ km | 345 |
| | | | | $\ell_{r,t} \sim U(500, 5000)$ yr | 3254 |
| | | | | $\sigma_l \sim U(0.1, 3.3)$ m | 0.14 |
| | | | | $\ell_{l,x} \sim U(64, 319)$ km | 317.4 |
| | | | | $\ell_{l,t} \sim U(100, 2000)$ yr | 1978 |
| US Atlantic spatio-temporal analysis | Empirical Bayesian; NI | Normal likelihood with additional white noise | Gaussian Process with ICE_7G as mean function and multiple isotropic kernels | $\omega \sim U(0.01, 10)$ m | 0.02 |
| | | | | $\sigma_r \sim U(0.2, 10)$ m | 6.1 |
| | | | | $\ell_{r,x} \sim U(319, 1593)$ km | 1586 |
| | | | | $\ell_{r,t} \sim U(500, 5000)$ yr | 4683 |
| | | | | $\sigma_l \sim U(0.1, 3.3)$ m | 0.12 |
| | | | | $\ell_{l,x} \sim U(64, 319)$ km | 312 |
| | | | | $\ell_{l,t} \sim U(100, 2000)$ yr | 1970 |
| US Atlantic spatio-temporal analysis | Empirical Bayesian; NI | Normal likelihood with additional white noise | Gaussian Process with zero mean and a sampling kernel determined by a GIA model ensemble | $\omega \sim U(0.01, 10)$ m | 0.4 |
| US Atlantic spatio-temporal analysis | Variational Bayesian; NI | Normal likelihood with additional white noise | A GIA model ensemble (consist of 9 individual models) | $\omega \sim U(0.01, 0.5)$ | 0.042 (0.028, 0.056) |
| | | | | $\nu_1 \sim Beta(0.22, 0.78)$ | 0.04 (0.03, 0.05) |
| | | | | $\nu_2 \sim Beta(0.02, 0.98)$ | 0. (0., 0.) |
| | | | | $\nu_3 \sim Beta(0.02, 0.98)$ | 0. (0., 0.) |
| | | | | $\nu_4 \sim Beta(0.09, 0.91)$ | 0. (0., 0.) |
| | | | | $\nu_5 \sim Beta(0.01, 0.99)$ | 0. (0., 0.) |
| | | | | $\nu_6 \sim Beta(0.02, 0.98)$ | 0. (0., 0.) |
| | | | | $\nu_7 \sim Beta(0.16, 0.84)$ | 0.39 (0.36, 0.42) |
| | | | | $\nu_8 \sim Beta(0.29, 0.71)$ | 0.35 (0.32, 0.37) |
| | | | | $\nu_9 \sim Beta(0.17, 0.83)$ | 0.22 (0.20, 0.24) |

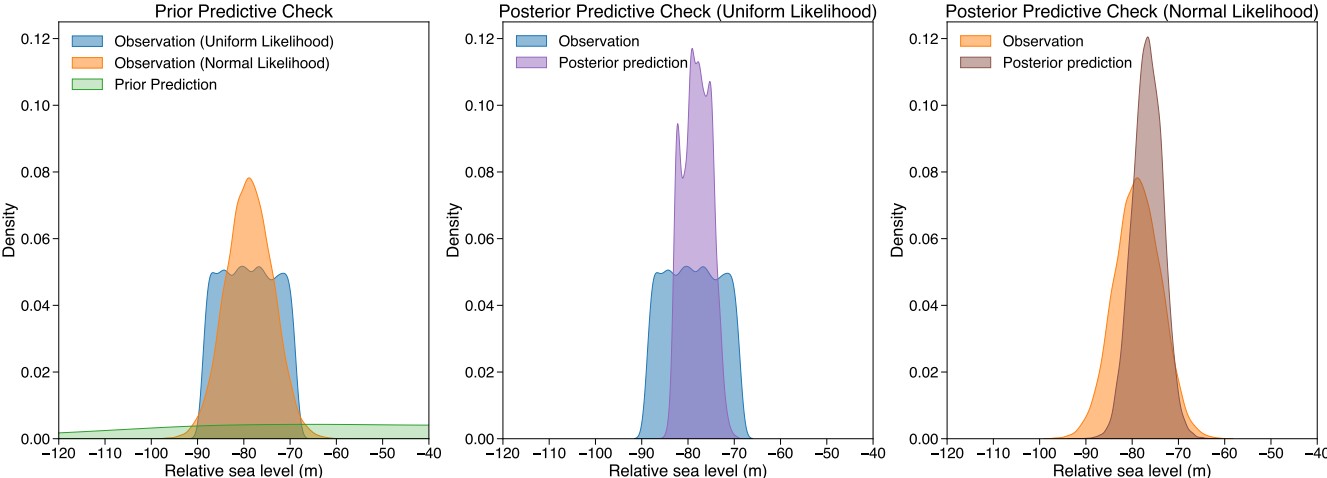

**Figure A1.** Prior and Posterior Predictive Checks for modeling results presented in the main text, section 4.1.1. A random data point was selected for illustrative purposes. Left, Prior predictions compared with observational data, assuming uniform (blue) and normal (orange) likelihood functions. Middle, Posterior predictions compared with observational data, assuming a uniform likelihood function. Right, Posterior predictions compared with observational data, assuming a normal likelihood function.

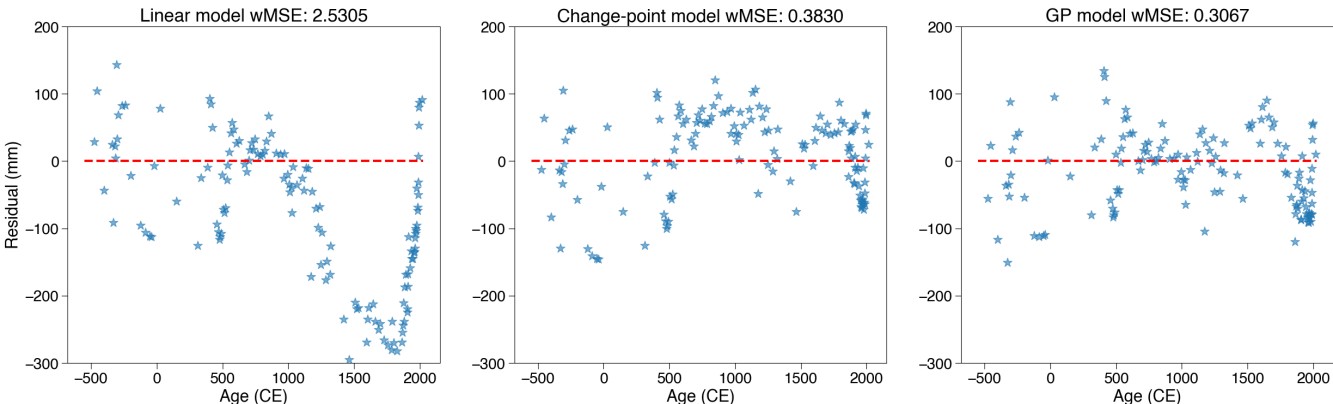

**Figure A2.** Residual plots for three process level model introduced in main text, section 4.1.2. The weighted mean square error (wMSE) for each model is given as figure titles. Red dashed line represents 0 error.

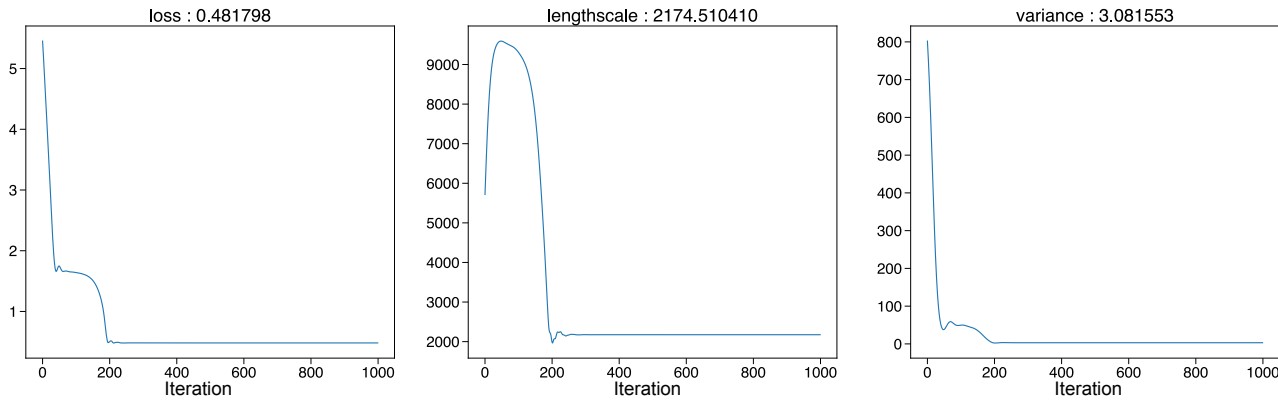

**Figure A3.** Optimization trace plot of empirical Bayesian analysis introduced in main text, section 4.1.3.

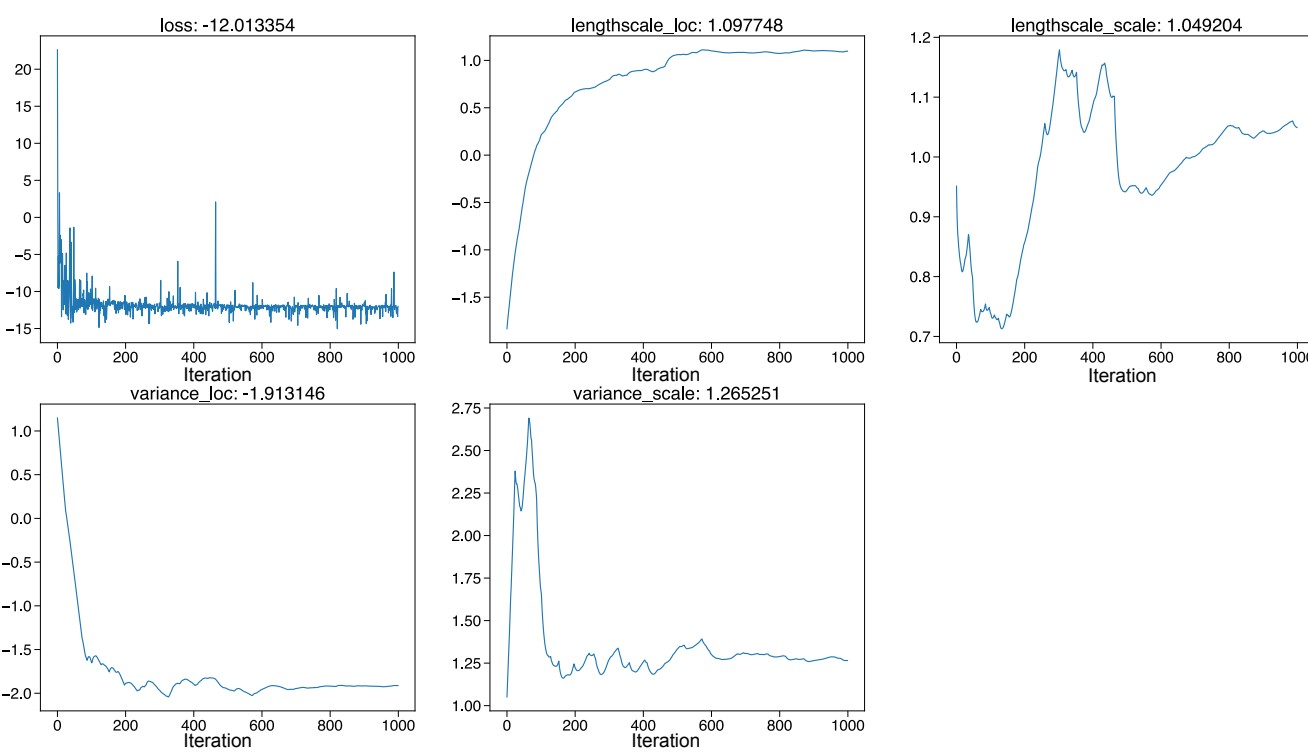

**Figure A4.** Optimization trace plot of Variational Bayesian analysis introduced in main text, section 4.1.3. Note that the parameters shown here are variational inference parameters in Pyro, which are optimized to approximate the posterior distribution but do not directly correspond to the actual parameters of probability distribution.

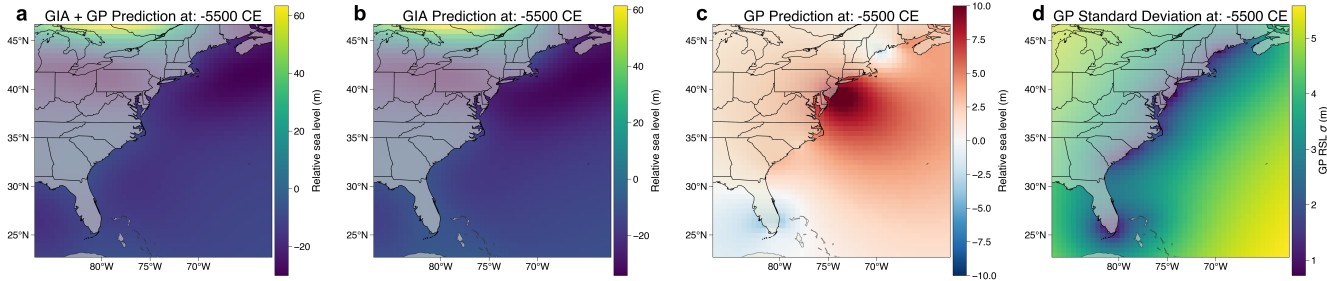

**Figure A5.** Model prediction from model *ii* for relative sea level at time point -5500 CE along the US Atlantic coast. (a) Mean relative sea-level prediction,representing the sum of components (b) and (c). (b) Relative sea-level prediction derived from the Gaussian Process mean function, based on a glacial isostatic adjustment model incorporating the ICE_7G ice model and VM5a Earth model (Roy and Peltier, 2018; Peltier et al., 2015). (c) Relative sea-level change induced by the GP covariance function. (d) Standard deviation of relative sea level, as estimated by the GP covariance function.