# Peer review of "PaleoSTeHM v1.0: a modern, scalable spatio-temporal hierarchical modeling framework for paleo-environmental data"

_EGUsphere, 2024_

## Referee Comment (RC1)

**Paper Review: PaleoSTeHM v1.0-rc: a modern, scalable spatio-temporal hierarchical modeling framework for paleo-environmental data**

November 2024

**Summary of content**

The authors present a novel spatio-temporal hierarchical modeling framework designed for examining paleo-environmental data. It provides an in-depth discussion of the underlying architecture of the PaleoSTeHM software and showcases its capabilities through several case studies focused on paleo sea-level data.

**Comment to the Author**

This paper showcases the PaleoSTeHM software, which represents a significant and valuable contribution to the field. The integration of machine learning techniques with a variety of Bayesian inference methods marks a notable advancement. However, the paper's structure, terminology and layout would benefit from further refinement. It assumes considerable prior knowledge, which may pose challenges for readers. I have outlined several questions and observations regarding the explanations, along with substantial content-related feedback (refer to Main and Minor comments).

**Main Comments**

- The target audience for this paper is somewhat unclear. While the introductory section provides an overview of statistical methods, the machine learning component and the software structure is not explained in sufficient detail. Additionally, the introduction feels incomplete and the paper's layout inconsistently transitions between examples, such as those involving paleo-sea level data. Furthermore, the paper assumes prior familiarity with Ashe et al. (2019), particularly regarding the authors' conventions for distinguishing between analytical choices (e.g., Variational Bayes) and modeling choices(e.g. GP). This assumption could make it challenging for readers unfamiliar with the referenced work to fully understand these distinctions. I would recommend keeping consistency with terminology throughout the paper. I would ensure the paper focuses on the capabilities of the software.

- The paper lacks a dedicated section on model validation, and I could not find any methods addressing this in the GitHub tutorial. How can users assess and ensure model convergence? Were prior posterior predictive checks performed or was a simulation analysis conducted to evaluate the techniques? Additionally, were cross-validation methods employed, with residual analysis and true versus predicted plots? It is also unclear how users should compare models—for example, using metrics like RMSE, MAE or empirical convergence. I strongly recommend including a comprehensive section on model validation in the paper, along with links to resources or tutorials demonstrating how these assessments can be performed using the software.

- The literature review on statistical models lacks references that directly link to the introductory equations, which undermines the connection between the theoretical framework and existing research. Additionally, the paper provides extensive details on a wide range of topics, from physical models to various proxy data sources. Please ensure that appropriate references are included for all these topics. For instance, I came across a recent paper by Upton et al. (2024) that used Bayesian spatio-temporal generalized additive models and an R package available on Cran called reslr, which appear to adopt a similar approach to modeling sea-level changes. A comparison of the methods employed in the current work and those in reslr and similar approaches would be particularly valuable for readers.

**Minor Comments**

- Title: "for paleo-environmental data". The paper primarily focuses on paleo-sea level data, and the tutorials associated with the software exclusively use this dataset. It would be helpful if the authors included an example in the GitHub repository demonstrating how the methodology could be adapted for other types of proxy data, such as temperature. Alternatively, if the scope of the paper is limited to sea-level data, the title might be revised to reflect this focus more accurately. If I have overlooked another example where an alternative climate proxy is examined, it would be beneficial for the authors to highlight it more clearly.

- Update ", and " through out the paper to remove the comma before the "and".

- Abstract:
  - "Geological records of past environmental change provide crucial information for assessing long-term climate variability, non-stationarity, and nonlinearities" in climate? This sentences seems unfinished.
  - "This framework enables the implementation of flexible statistical models that rigorously quantify spatial and temporal variability from geological data with clear distinguishing between measurement and inferential uncertainty from process variability".. An improvement: "clearly distinguishing measurement and inferential". Additionally, some sentences in the paper could be shortened to improve readability and clarity.

- Introduction:
  - Line 11: "As humans push the planet's climate and biosphere increasingly far outside the range of our species' experience, the geological record provides critical out-of-sample data against which to test the models used to project future environmental change". This sentence is misleading as the paper does not address projections.
  - " Yet, as an environmental record, the geological data is quite sparse and often noisy and indirect." Revision: "However, as an environmental record, geological data is sparse, often noisy, and indirect." Also, there is minimal discussion on what the geological data is. Include more information about the data or use a reference to direct readers to e.g. Shennan et al 2015 for paleo-sea level data.
  - Line 16: "an analytical perspective" keep the naming convention consistent. In Table 1 you use analysis choice and other locations interchange these terms.
  - Line 20: Include a definition of a geological proxy and references for examples of "temperature and precipitation".
  - Line 23 and Line 24: the acronym GMSL needs to be defined.
  - Line 27: "(Tan et al., ...)" include "e.g." at the beginning of this list as it is not the complete list of papers.

- Hierarchical statistical Modeling:
  - Providing a definition of hierarchical modeling at the start of this section would help readers understand the topic more effectively.
  - I would begin by explaining Bayesian statistics and then link it to the example you will describe in the next section. Include references to the original mathematical papers for the Bayesian and conditional probability definitions.
  - Line 40: " data (y) can be inverted". I would use a different verb then inverted as this is misleading.
  - Table 1: I recognize that the table is to reflect the table in Ashe et al. 2019, yet, there are a number of relevant terms used extensively in the paper which would be beneficial if they were included in this table, e.g. Variational Bayes, Bayesian statistics, empirical Bayes, full Bayes, machine learning terms. Highlight where these terms are used throughout the paper, i.e. refer to Table 1.
  - Line 53: " A basic hierarchical statistical model for paleo sea level distinguishes the fundamental RSL change from both its inherent variability and the observational noise." I would place this sentence after equation 3 and clearly state that is the your example case. This is an example where the layout of the paper is blending the example with definitions as mentioned in the first Main Comment. Also, RSL acronym has not be defined. Explain to the reader what is relative sea level.
  - Line 61: Missing references for "geochronology techniques", for example Wright et al 2017.
  - Line 67: Include GIA acronym with the definition of glacial isostatic adjustment. Also, Table 1 has been referenced here but does not correspond to any of the terms in the table.

- Line 72 and 73 are very important as they define the authors' conventional terminology i.e. modeling choices and analysis choices. As mentioned in the Main comments section, this is where the authors need to clarify their meaning of analysis choice and model choice before describing how this applies to the new software.

- Line 74 and 75 highlights how prior knowledge has been assumed as mentioned in the first Main comment. The authors need to include brief explanations for deterministic methods and the difference between those methods and probabilistic methods.

- "Several factors, including the complexity of the problem, the size and resolution of the data available, the computational resources at hand, and the extent of prior knowledge applicable to the modeling effort, should guide the selection of modeling and analytical choices." Shorten this sentence to improve readability. This is a key sentence for future software users, again relates to the first Main comment.

- Model Description: Would this benefit with an update of name from model description to Software Architecture? This is the key section of this paper.

  - Section 3.1 needs to be restructured. The authors use L3, L2, and then L1 without defining these modules. I would recommend including lines 95 to 100 earlier in the section for clarity.

  - Line 89 and 90 "L2 employs Python as the user interface language and utilizes a high-performance machine learning platform as the execution back-end". This sentences needs more explanation.

  - Line 96: "including auto-differentiation, GPU acceleration, and modern optimization algorithms." Include definitions and references for each of these topics.

  - Line 100: " multiple methods to consider temporal uncertainty". Include how the uncertainty in the response is included. Also, address how each model fits into Figure 1. Instead of placing Figure 1 in brackets at the end of a sentence, consider rephrasing to begin the sentence with a reference to the figure. For example: "As shown in Figure 1, L3..." This approach integrates the figure more seamlessly into the narrative and emphasizes its relevance to the discussion.

  - Line 104: Should this not be "(L4, Figure 1)"? Or is this section describing how L3 interacts with L4?

  - Figure 1: The caption does not include the term module which has be extensively used in the section above. Include a sentence on how these modules or layers interact. There should be references used for Pyro and PyTorch in the caption. The discussion about L1 in both the caption and the corresponding section is very limited. Consider adding an additional sentence to the text to provide more context or explanation about L1, ensuring its significance is adequately addressed.

  - Section 3.1: "experiment architecture" does this relate to the result section or is this section address how the user should use the software?

  - Line 112: "Training" this term is confusing. Will the user be repeating these steps " sequential selection steps" or is it that they are identifying the best option for their data?

  - Line 113: This relates back to the first Main comment. Instead I recommend that the authors start with: " In Figure 2 we define the 5 steps of the PaleoSTeHM software focusing on L3 from Figure 1. These steps are...

  - Line 116: " These five steps reflect core functionalities developed within three PaleoSTeHM modules, shown in Figure 1." Should this not be " the core functionalities of Layer L3 from Figure 1. This needs clarity.

  - Line 117: "To support the effective selection of modeling and analytical choices provided by PaleoSTeHM for various paleo-environmental applications." This sentence is unfinished.

  - Line 118: "modeling option" update to " modeling choices" for consistency.

  - Figure 2 is a very important figure and the caption needs to discuss how it relates to Figure 1. Again, "experiment architecture" is a confusing term. I would number the 5 steps, instead of using colors and boxes to define the steps as this is not inclusive for all readers. "temporal uncertainty treatment" is not defined anywhere, should this reference the EIV method (Dey et al 2000) and the NI method ( and McHutchon and Rasmussen, 2011). Include e.g before "temporally linear and Gaussian Process"

  - Line 122: "commonly used temporal or spatio-temporal modeling choices used". Update this sentence.

  - Line 125: "While we do not include a specific section for parameter-level modeling, leveraging the ecosystem of Pyro and Pytorch enables users to easily define prior probabilities for data and process-level model parameters using most of the commonly used probability distributions". Include references for Pyro and Pytorch. Additionally, improve the readability of this sentence. Does this mean that the software allows users to define priors?

- "e.g., radiocarbon Reimer". should this include "radiocarbon dating, Reimer.."
- Could equation 4 not be discussed in section called hierarchical statistical modeling?
- Line 145 and 146: Explain why strong covariance could be an issue and how adapting the likelihood structure could improve this.
- Lines 150 and 151: The term "likelihood sampling code" has not been defined, which may confuse readers. Please provide a clear explanation of what this means and how it is used in the context of the model. Link this back to the discussion in Line 146. Additionally, Pyro is mentioned without a reference—please include the relevant citation.
- The "Process Level Modeling" section should be restructured as a new subsection, with the subsequent sections organized as subtopics within it, i.e. temporal linear models to GP Kernal module.
- Line 160: "qualitatively assessed". How was this carried out? Was a residual analysis undertaken? Include a reference with this statement.
- Equation 8 and the other equations in this section should be kept general to apply to various paleo-environmental changes. For example, $\beta$ could represent the rate of change in the paleo-environmental variable. Additionally, the authors should reference Ashe et al 2019 for the upcoming sections.
- Include reference for change point Carlin et al 1992.
- Line 176: define non-parametric and parametric in the table.
- Line 196:" Yet, its justification can sometimes be complex (Stein, 2012)." Expand.
- Line 203 and 204: Should include a reference.
- Line 211 missing key reference to Peltier's multiple GIA models.
- Line 213: "spatial teleconnections" update this term as it is misleading.
- Line 217: " PaleoSTeHM to probabilistically..." expand on this feature. This is very important and a novel component of the software.
- Line 221: given a definition for "sampling covariance functions".
- Line 225: Figure 2 instead of Figure 1?
- Line 226 to 229: The list requires a reference to the original statistical paper where the technique is defined, along with the examples of where it is used in the paleo-sea level field.
- Line 235 Include reference.
- Line 237-238: "temporal and spatial white noise kernels". Need to expand on this and include references to equation 4.
- Line 243: "The spatial correlation is computed for spatial kernels based on the 1-dimensional geographical radial distance between data points." Expand on this.
- Table 2: Please include references to the statistical papers where these equations were defined.
- Analysis Choice and Modeling Choice. It is confusing when the analysis choice which represents least square or Bayesian approaches keeps changing its name. Similarly modeling choice or model characteristic keeps changing terminology.
- Line 249: "deterministic methods (e.g..." This has not been explained previously or in Table 1. Also "implemented in other packages", reference the other packages?
- Line 260-265: "sampling process", "autocorrelation", "effective sample size per iteration and..", "path length" and "step size". All require definitions.
- Line 283: "details below": Include the specific section.
- Line 284-287: Could you provide a more detailed explanation? The current text assumes prior knowledge of many machine learning techniques and terms.
- Line 296: Need to include a reference for variational bayes.
- Equation 12 needs more explanation and Kullback-Leibler divergence needs a reference.
- Line 302-303: "..scales linearly". Is there a reference for this?
- Line 308: "Cahill.." missing reference to "Dey et al, 2000:.

- Results: Consider renaming this section to Case Studies. Additionally, the layout should more closely align with the steps outlined in Figure 2 and Section 3.2.

  - Line 319 include link to codes.

- Line 320: Update the sentences into bullet points of the different test cases. References for the data sources are missing. Also, "analysis choice modeling techniques" blending the two separate modules, update this.

- Line 325: should there be more references include e.g. Walker 2021?

- Line 330: Have a sentence at the start stating what is being discussed, i.e. "in this section we will address the impact of data level". Also, there is minimal discussion on what the input data is.

- Equation 14 and 15: The treatment of the white noise component in both equations is unclear. Is the white noise modeled as a random variable drawn from a specified distribution, or is it treated as a deterministic fixed parameter? This distinction needs to be clarified. Additionally, the notation in Equation 15 is non-standard, as the inclusion of the square root means that the second parameter no longer represents the variance. Please clarify whether the square root is intentional and, if so, provide an explanation for this modeling choice.

- Figure 3: The caption for Figure 3 requires more detail. Please reference the data source used in the figure. Clarify what the additional noise component represents, is it the standard deviation of the white noise or the residual standard deviation of the model? Also, explain why 90% credible intervals are used and why uncertainty boxes are set to 2 $\sigma$? Was a 2 *sigma* uncertainty applied in the models, or was it 1 $\sigma$? Add labels (a), (b), etc., to each panel for clarity. The legend is missing for the RSL vs. Age plot. For readers unfamiliar with RSL, include a brief explanation of why RSL values are negative.

- Line 347: How does the user determine whether to use a normal or a uniform distribution? The results show a difference between the two, but is there a preferred distribution, or should the choice depend on the specific characteristics of the data?

- Line 355: "3 change points". Why 3 change points? Should you reference the original paper? Also, "RBF" acronym not defined.

- Figure 4: Update RSL with Relative Sea Level. Include legend explaining the red boxes and in caption describe how you model the midpoint of the boxes with 1 $\sigma$ error. Include reference for the data. Label each row of plots (a). Why was variational Bayes not used for GP in time?

- Line 370: Why not use the same data location throughout the case study in order for the reader to clear see the impacts of the model and analytic choices on the same dataset? "use a subset of original data" is this to speed up the model run times. How many data points were used?

- Line 372-375: Include the run times in this paragraph as stated in Figure 5. Is 2200 posterior samples and 500 iterations sufficient, or does this seem on the lower end as the default is 5000 for other software? Additionally, could the model convergence checks for all approaches be included in the appendix for consistency?

- Figure 5: Include reference for data. Update RSL with relative sea level and include legends explaining the red boxes. Label each row for easier explanations.

- Section 4.2: Spatio-Temporal Analysis: This section requires further refinement to enhance readability and clarity. It lacks sufficient references to the original data collection process and previous model results, which are crucial for context. Additionally, the model definitions assume a level of prior knowledge that may not be accessible to all readers. The discussion of Figure 6 is also lacking, as the individual panels have not been thoroughly addressed. Each panel should be analyzed in detail or with a brief summary, comparing the results from the different model choices after discussing them independently.

- Line 390: This section examines the spatio-temporal model used to examine the different RSL drivers

- Line 392: Include link to the sea-level proxy database.

- Line 395 - 400: Present this paragraph as a table and name the models instead of using (i). With the corresponding equations. Describe the purpose of each of these models.

- Equations 16 - 21: The paper encourages the reader to refer to Ashe et al 2019, however, the notation does not correspond completely. For example, "g(t)" is not described as the "global component" in the text and Line 416 " g(t) as we assume..." expand on this. Also, all the $K$s should be defined or reference to Table 2.

- Lines 423-430: The terms "sampling covariance function" and "physical ensemble m" require further explanation and appropriate references. This section needs be improved. For example: "Lin et al. (2023b) described a method for incorporating physics-based GIA models using an ensemble approach. In our software, the ensemble method is implemented as...". Please clarify these concepts and provide the necessary references to improve the readability and context of this section.

- Line 431: Provide an explanation for " weighted mean of different physical models" including a reference. Similarly Line 435 - 440 assume prior knowledge and do not include references, this needs to be altered.

- Line 440 - 454: Update " Figure 6 demonstrates the results from our 5 spatio-temporal process level models defined in Table (3)". I would recommend reviewing the layout of this paragraph to improve readability. For example: "regional common kernal, g(t), in equation 17" has not been clearly defined. Include a recap of what model (i) is examining. Check references, for example Cahill et al., 2015 did not examine data in Florida. Describe each panel of Figure 6 in a specific order, for example "Figure 6i, geological sea-level..", is this (i) representing the model or is it representing a different panel plot that is highlighting a specific process?

- Line 455: "GIA model" missing a reference.

- Line 464: "teleconnections"? What is this referring to?

- Figure 6: Include legend to show what the red box is. Include labels for each row explaining what the panels represent. Update model i - iv with the name of the models used. Caption: "Process level models impact.." instead " The impact of process level modeling choices for..". Include reference to the data. The labels a - l are hard to read. Also we now have model i and panel i. Reference for ICE7G is required. Why examine the year -5500CE? Explain the standard deviation of RSL prediction more clearly.

- Line 471: "is equivalent to a linear combination of physical models according to data-model misfits". Clarify this sentence.

- Line 477- 478: Either include a reference or explain what is meant by " direct constraints on ice history, the ..."

- Discussion:

  - Line 481 - 485: Improve the readability of this sentence. A paragraph should contain a least 3 sentences and these sentences are long.

  - Line 486: "Because of" avoid using because at the start of any sentence.

  - Line 488: " process level models introduced this paper" .. "in this paper".

  - Line 490: "to describe latent some space-time..". Improve this sentence.

  - Lines 495-498: The term "principal component..." requires either references to alternative methods or a clear definition of what these methods are. Please provide the necessary context for clarity.

  - Line 500: Upton 2023 used generalized additive models for RSL changes. This should be included.

  - Line 506-510: "subject to complex likelihoods". This requires a reference.

  - Line 520: " Another outstanding issue for GP based process level models is scalability, the standard GP models included in PaleoSTeHM v1.0 cannot scale well to large data sets (>10 thousands data points)." Has there been test done to examine ¿10 thousands or 10 thousand data points? More common to describe the computational requirement of a Gaussian Process being of $O(n^3)$ where n is the number of data points.

  - Line 530: "and (Lin et al., 2023b) developed" remove the brackets.

- Conclusions:

  - Line 545: "though the limited availability of user-friendly software often hinders it." There are other packages available for the paleo-sea level community which have not been referenced in this paper.

  - Appendix, Table A1: Include more information in the Appendix regarding its relevance to the paper. Why was 90% credible interval used in this paper? Convention is 95%, is there a reason why 90% was used instead? Should parameter level title include "Priors for Parameters". Need to define GBR in caption for table. Should you include reference to where the data is sourced? Also a reference to where the models have been used in the past?

  - Appendix Figure A1: Update "Model ii" with the name of the modeling option and analysis choice which has been used. Improve text in caption "prediction on - 5500CE RSL", for example " Model predictions from Model X for relative sea level at time point -5500CE...". Include more information about how this plot relates to the paper and its relevance. Included a reference to highlight the data source. Linked in the paper on line 455, however it requires more discussion e.g. (as shown in Figure A1). The caption needs more information regarding the axis. RSL should be Relative Sea Level. The reference for the ICE model and VM model should be included in the caption. Could the letters a,b,... be placed outside the plots in black to make it easier to read.

- I commend the author's Github repository which contains many tutorials that demonstrate to the user how to implement the PaleoSTeHM. I recommend reviewing the documentation to improve readability, there is a number of spelling mistakes and some of the sentences are long and misleading. Additionally, the software

requires a google drive connection is there another option as some university do not allow Google accounts? The 2 hour tutorial videos are very useful however, is there any possibility to split them into smaller segments in order to be used in future lectures.

---

## Referee Comment (RC2)

This manuscript presents an overview of open source spatio-temporal modelling software aimed at the paleo-environmental community. The modelling approaches are formulated in a hierachical Bayesian context with different implementations sampling algorithms of varying complexity (and calculation speed) and these are demonstrated with a variety of case studies (limited here to sea-level models). Overall, the idea of putting these tools out there is to be welcomed, and I feel this will be a useful contribution in terms of advertising the availability of the software. In the end, success will be determined by the uptake by non-expert users. To this end, at least from my experience, key aspects are good documentation, relative ease of use and some tutorial examples that allow a user to repeat, and develop on, previous studies. Given the nature of the methods implemented, there is a degree of technical know how required to understand the statistical details but as far as I can see, the examples are all available and should allow a new user to at least try things out without perhaps necessarily being on top of the details. So, the manuscript needs a bit of work (minor changes), but I think this is worth publishing in the sense of put it out there and see how it goes - hopefully well and ideally leading to a wider range of applications than presented here.

**Some points**

I felt there are perhaps some technical terms/acronyms that could be defined for the less experience potential users. Too many undefined terms and/or jargon will tend to put off the non-expert and/or those not in the sea-level community – I have highlighted some below. In addition to the definition of terms table... perhaps a table defining the symbols, etc used may be use.

There is no mention of transdimensional MCMC methods, nor model choice (in terms of model complexity for example)...the latter is not alawys simple but is there any sense of this in the current version (or planned for future versions ?)

L13 out-of-sample data ??? Not sure what this means.

L14 (and L45) for me data are plural and should use are not is (data are), but this may be a slightly old fashioned view.

L17 do we mean reconstructing paleo-environment or paleo-environmental signals ?

L18 – perhaps define hierachical here...

L19 clearer (than what ?)

L20 – It has been...you mean hierachcial statistical models from the previous sentence...They have been ?

L23 GMSL – Global Mean Sea Level...

L26 I would start a new paragraph after 2013..

L26 such techniques ? Remind us in a word or two.

L45 Marginal likelihood, or evidence...(could define it  $p(y) = \int p(y|\theta)p(\theta) d\theta$ )

Perhaps the phrase static observations is a little ambiguous (confusion with temporal data)...would it be better to say a given data set (which could include temporal data).

Table 1

Hyperparameter – parameter of a prior distribution (to be inferred) ? MCMC – random sampling (need a space) Residuals – definition implies this should not be plural.

L53 RSL – Relative Sea level ?

L55 inverting conditional probabilities – not sure we formally inverting conditional probabilities, but use Bayes' theorem (historically this was often considered as the inverse probability problem).

L58 Eqn 3 – not too important, but the last term could be written as the product of the two probabilities as  $\theta s$  and  $\theta d$  are being treated as independent ?

L66 Conditional model parameters ? Also could the unobserved physical parameters being treated as nuisance parameters (that we need to know, but are not that interested in...but perhaps here we can not integrate them out).

L68 linear rate in a linear sea-level ..constant rate (too many linears ?)

L94 Paszke et al. 2017...perhaps this the best citation for PyTorch ?

L104 L3 should be L4 ?

Eqn 4 perhaps just redefine y here as observed data (or prediction ...as uncertainty is defined as difference in prediction and true value).

L140 – for me the covariance matrix contains the errors covariance, not data covariance and the common assumption of cov = 0 is that the errors are not correlated...this is not always the case in time series (as implied in the text). This seems to be stated later..L238 – uncorrelated uncertainty ?

See also Sambridge, (2016), Reconstructing time series and their uncertainty from

observations with universal noise, J. Geophys. Res. Solid Earth, 121, 4990–5012,

doi:10.1002/2016JB012901 and/or Bodin et al. (2012), Transdimensional inversion of receiver functions and surface wave dispersion, J. Geophys. Res., 117, B02301, doi:10.1029/2011JB008560.

L145 Why Whereas ? ... Perhaps in practice ?

L153 paleo-env. signal is modeled ?

L158 for me a linear model is not just a straight line, it is where the forward problem is linear, such that model parameters can be separated from a kernel matrix (so a polynomial of order 20 is still linear)....perhaps just say straight line model

L165 – what happens if we do not know m, the number opf changepoints ? any comments on transdimensional change point models (or indeed transdimensional models in general)..e.g. Gallagher et al. (2011), Inference of abrupt changes in noisy geochemical records using Bayesian Transdimensional changepoint models, Earth Planet. Sci. Lett., 311, 182–194.

L173 – not clear to me...the product of  $\beta_{k-1}$  and the difference of  $\gamma_k$  and  $\gamma_{k-1}$ ....what does this mean in practice ?

L181 – covariance matrix - data or data errors ?

L196 why Yet ?

L205 fundamental (not foundational)..

L213 - will allow ... should allow ?

Table 2 caption t denotes age of the data...a little clumsy ...could be the age of the analysis...a few months. Age of the sample ?

L253 – again a little clumsy...to take a range of probable values....to take on a range of values,

potentially with different probabilities

L267 – why do you say relative (explain this...).

L282 -283 — not clear for me...explicit prior for the parameters (distinguish from hyperparameters, if relevant, with an example ?)

L284 optimize hyperparameters using their maximum likelihood estimates..

L318 – to demonstrate (rather than enchance)...

L330 – we can look at residuals to assess if our model produces normal distributed uncertainties ?

L345 – noticeable, but significant? Does not really look too important ij terms of the uncertainty ranges on the slope.

Fig 3 – can we explain why the uniform likelihood gives multimodal distributions on the rate and intercept ?

How do the correlations of these two parameters differ for the two likelihoods...?

L355 – could we let the number of changepoints vary - and the same for fig 4 – if we had 10 changepoints...perhaps we capture the GP model form better ?

L336 – language a little clumsy – through a variational Bayesian maner...rephrase.

Fig 4 – are the likelihoods much different for the preferred model and for the MCMC samplers can we see the likelihood as a function of iteration ?

L366 – environmental statistics problems ? How do we know the model is too smooth ? L378 their underlying uncertainty.

L383 bias of the perhaps overly simple approach?

L390 no need for The...

Fig 5 tge bottom rifght panel is a little hard to read with the different colours overlapping...I could not figure out where the orange region is ...you could trace the bounding curves to show that

L396 define GIA ?

L398 ensemble

L409 their prior distributions

L415 mean expectation of what ?

L416 – mean = expected ?

L472 weighted according ?

L482 – something missing between hierachical model and any statistical model

L499 – comma rather than and before Piecuch

L549 In contrast to what?

---

## Author Response (AR1)

Department of Earth and Planetary Sciences
Rutgers University – New Brunswick
Piscataway
USA
08854

email: yc.lin@rutgers.edu

26 Jan 2025

Dear Dr Phipps,

I am pleased to submit the revised manuscript titled *PaleoSTeHM: A Spatio-Temporal Modeling Framework for Paleo-Environmental Studies* for consideration in *Geoscientific Model Development (GMD)*. We greatly appreciate the reviewers' constructive feedback, which has helped us improve the manuscript significantly.

Major revisions include:

1. **Target Audience and Readability**: The introduction and terminology were refined, with key definitions added to Table 1 for accessibility.
2. **Model Validation**: A new section (3.6) synthesizes validation methods like convergence diagnostics and residual analysis.
3. **Expanded Context**: The literature review now includes comparisons with similar tools (e.g., *reslr* by Upton et al., 2024).
4. **Enhanced Documentation**: GitHub tutorials were refined, and video resources split into smaller, more accessible segments.

We believe these revisions address all reviewer concerns and have greatly improved the manuscript's clarity, depth, and usability.

Thank you for considering our submission.

Yours sincerely,
Yucheng Lin on behalf of Robert Kopp, and other co-authors

**Review 1**

This paper showcases the PaleoSTeHM software, which represents a significant and valuable contribution to the field. The integration of machine learning techniques with a variety of Bayesian inference methods marks a notable advancement. However, the paper's structure, terminology and layout would benefit from further refinement. It assumes considerable prior knowledge, which may pose challenges for readers. I have outlined several questions and observations regarding the explanations, along with substantial content-related feedback (refer to Main and Minor comments).

We thank the reviewer for recognizing the importance and thoroughness of this study, as well as for acknowledging the effort invested in developing the PaleoSTeHM framework. We also greatly appreciate the reviewer's detailed comments and constructive feedback that significantly improve the quality of this paper. We have addressed all comments in detail below, as highlighted in blue.

**Main Comments**

- The target audience for this paper is somewhat unclear. While the introductory section provides an overview of statistical methods, the machine learning component and the software structure is not explained in sufficient detail. Additionally, the introduction feels incomplete and the paper's layout inconsistently transitions between examples, such as those involving paleo-sea level data. Furthermore, the paper assumes prior familiarity with Ashe et al. (2019), particularly regarding the authors' conventions for distinguishing between analytical choices (e.g., Variational Bayes) and modeling choices(e.g. GP). This assumption could make it challenging for readers unfamiliar with the referenced work to fully understand these distinctions. I would recommend keeping consistency with terminology throughout the paper. I would ensure the paper focuses on the capabilities of the software.

  We thank the reviewer for highlighting the importance of clarifying the target audience and improving the readability of the paper. We have now revised the manuscript to address these concerns. Specifically, we have introduced essential definitions either in Table 1 or within the main text to ensure readers are equipped with the necessary context. Additionally, we have standardized the terminology throughout the manuscript, using consistent language for analysis choices and modeling choices to avoid confusion (see detailed response to each point below). Finally, we have focused on discussing the capabilities of the PaleoSTeHM software, ensuring that the paper highlights its core features and practical applications. We believe these changes will make the paper more accessible and comprehensible to a wider audience.

- The paper lacks a dedicated section on model validation, and I could not find any methods addressing this in the GitHub tutorial. How can users assess and ensure model convergence? Were prior posterior predictive checks performed or was a simulation analysis conducted to evaluate the techniques? Additionally, were cross validation methods employed, with residual analysis and true versus predicted plots? It is also unclear how users should compare models—for example, using metrics like RMSE, MAE or empirical convergence. I strongly recommend including a comprehensive section on model validation in the paper, along with links to resources or tutorials demonstrating how these assessments can be performed using the software.

  We thank the reviewer for highlighting this critical aspect of our work. While we previously included some model validation test information in our GitHub tutorials, such as residual plots with weighted mean square error (wMSE) for every process-level model (Tutorial 2), effective sample size and Gelman-Rubin statistics for MCMC-based methods (Tutorial 3), simulation analysis for temporal Gaussian Processes (Tutorial 4), and prior posterior predictive checks alongside true vs predicted plots for GP model (Tutorial 6), we acknowledge that our manuscript lacked a dedicated section synthesizing this information systematically.

  In response, we have now included an additional section 3.6 in the manuscript explicitly addressing model validation. This new section outlines the methodologies provided in the tutorials and offers further details on convergence diagnostics, residual analysis, simulation validation, and model comparison metrics (e.g., RMSE, MAE). Furthermore, we have revised each case studies manuscript to reflect this point.

- The literature review on statistical models lacks references that directly link to the introductory equations, which undermines the connection between the theoretical framework and existing research. Additionally, the paper provides extensive details on a wide range of topics, from physical models to various proxy data sources. Please ensure that appropriate references are included for all these topics. For instance, I came across a recent paper by Upton et al. (2024) that used Bayesian spatio-temporal generalized additive models and an R package available on Cran called reslr, which appear to adopt a similar approach to modeling sea-level changes. A comparison of the methods employed in the current work and those in reslr and similar approaches would be particularly valuable for readers.

  We thank the reviewer for highlighting the importance of linking the literature review to the theoretical framework and including references to recent developments in similar modeling approaches. While this paper primarily focuses on the development and functionality of the PaleoSTeHM software rather than serving as a comprehensive review, we have now included the recent study by Upton et al. (2024) in the manuscript. Additionally, we acknowledge the value of comparing PaleoSTeHM with similar tools. A comparative analysis between PaleoSTeHM and the reslr package, led by Fangyi Tan from Nanyang Technological University, demonstrates that both tools produce similar decomposition results for the Holocene Singapore database. This comparison will be made available online soon. An in-depth analysis of the performance of different software packages represents a valuable direction for future research and development. However, the primary focus of this paper remains on establishing a robust machine learning framework tailored for paleoenvironmental applications.

**Minor Comments**

- Title: "for paleo-environmental data". The paper primarily focuses on paleo-sea level data, and the tutorials associated with the software exclusively use this dataset. It would be helpful if the authors included an example in the GitHub repository demonstrating how the methodology could be adapted for other types of proxy data, such as temperature. Alternatively, if the scope of the paper is limited to sea-level data, the title might be revised to reflect this focus more accurately. If I have overlooked another example where an alternative climate proxy is examined, it would be beneficial for the authors to highlight it more clearly.

  We thank the reviewer to point out this important issue. We have now added a demonstration to model paleo temperature anomaly time series in our Github tutorial 2 (https://github.com/radical-collaboration/PaleoSTeHM/blob/main/Tutorials/1.Introduction/2.Process_level_modelling.ipynb). Also, in previous version, we did illustrate model paleo $CO_2$ concentration in our Gaussian Process tutorial. We hope those examples can demonstrate PaleoSTeHM's general ability for paleoenvironmental problems, instead of just paleo sea level.

- Update ", and " through out the paper to remove the comma before the "and".

  Modified.

- Abstract:

  - "Geological records of past environmental change provide crucial information for assessing long-term climate variability, non-stationarity, and nonlinearities" in climate? This sentences seems unfinished.

    This sentence has been re-written for better clarity.

  - "This framework enables the implementation of flexible statistical models that rigorously quantify spatial and temporal variability from geological data with clear distinguishing between measurement and inferential uncertainty from process variability".. An improvement: "clearly distinguishing measurement and inferential". Additionally, some sentences in the paper could be shortened to improve readability and clarity.

    We thank the reviewer for this helpful suggestion, we have now tried to shorten some sentences in this paper where appropriate.

- Introduction:

- Line 11: "As humans push the planet's climate and biosphere increasingly far outside the range of our species' experience, the geological record provides critical out-of-sample data against which to test the models used to project future environmental change". This sentence is misleading as the paper does not address projections.

  We thank the reviewer for this suggestion. We now clarified that the paper focuses on improving geological environment reconstructions, which can provide essential information to test physical models for projecting future environmental change, aligning with the paper's scope (improving paleoenvironment reconstruction).

- " Yet, as an environmental record, the geological data is quite sparse and often noisy and indirect." Revision: "However, as an environmental record, geological data is sparse, often noisy, and indirect." Also, there is minimal discussion on what the geological data is. Include more information about the data or use a reference to direct readers to e.g. Shennan et al 2015 for paleo-sea level data.

  Modified, additional reference included for paleo sea-level and temperature reconstruction.

- Line 16: "an analytical perspective" keep the naming convention consistent. In Table 1 you use analysis choice and other locations interchange these terms.

  We've changed 'analytical perspective' to 'modeling perspective' to maintain overall consistency.

- Line 20: Include a definition of a geological proxy and references for examples of "temperature and precipitation".

  Revised.

- Line 23 and Line 24: the acronym GMSL needs to be defined.

  Updated.

  Line 27: "(Tan et al., ...)" include "e.g." at the beginning of this list as it is not the complete list of papers.

  Amended as suggested.

- Hierarchical statistical Modeling:

  - Providing a definition of hierarchical modeling at the start of this section would help readers understand the topic more effectively.

    Adjusted as recommended.

  - I would begin by explaining Bayesian statistics and then link it to the example you will describe in the next section. Include references to the original mathematical papers for the Bayesian and conditional probability definitions.

    Bayesian statistics is now being introduced first, citation included.

  - Line 40: " data (y) can be inverted". I would use a different verb then inverted as this is misleading.

    Changed to 'derived'.

  - Table 1: I recognize that the table is to reflect the table in Ashe et al. 2019, yet, there are a number of relevant terms used extensively in the paper which would be beneficial if they were included in this table, e.g. Variational Bayes, Bayesian statistics, empirical Bayes, full Bayes, machine learning terms. Highlight where these terms are used throughout the paper, i.e. refer to Table 1.

    Updated as requested.

  - Line 53: " A basic hierarchical statistical model for paleo sea level distinguishes the fundamental RSL change from both its inherent variability and the observational noise." I would place this sentence after equation 3 and clearly state that is the your example case. This is an example where the layout of the paper is blending the example with definitions as mentioned in the first Main Comment. Also, RSL acronym has not be defined. Explain to the reader what is relative sea level.

    We thank the reviewer for pointing out the issue regarding the blending of definitions with examples, as well as the lack of clarity in defining the RSL acronym. We have revised the manuscript to ensure coherence by placing the mentioned sentence after Equation 3, clearly linking it to the example case. Additionally, we have provided a clear definition of relative sea level (RSL) in Table 1 to improve readability and ensure that readers are well-informed.

  - Line 61: Missing references for "geochronology techniques", for example Wright et al 2017.

    Added.

  - Line 67: Include GIA acronym with the definition of glacial isostatic adjustment. Also, Table 1 has been referenced here but does not correspond to any of the terms in the table.

> Thanks for finding this mistake, table and text have now been updated.

Line 72 and 73 are very important as they define the authors' conventional terminology i.e. modeling choices and analysis choices. As mentioned in the Main comments section, this is where the authors need to clarify their meaning of analysis choice and model choice before describing how this applies to the new software.

> We thank the reviewer for highlighting the importance of clarifying the distinction between modeling choices and analysis choices. To address this, we have revised manuscript to provide clear definitions and explanations for these terms. Specifically, we have elaborated on modeling choices as decisions related to the construction of the data, process, and parameter levels, while analysis choices pertain to the methodologies and algorithms used to implement and quantify uncertainty within the selected models. This distinction has been explicitly defined to ensure readers can readily understand its application within the PaleoSTeHM framework. We believe this clarification enhances the readability and interpretability of our manuscript.

– Line 74 and 75 highlights how prior knowledge has been assumed as mentioned in the first Main comment. The authors need to include brief explanations for deterministic methods and the difference between those methods and probabilistic methods.

> We thank the review to point out this problem, the definition about deterministic and probabilistic models are included now.

– "Several factors, including the complexity of the problem, the size and resolution of the data available, the computational resources at hand, and the extent of prior knowledge applicable to the modeling effort, should guide the selection of modeling and analytical choices." Shorten this sentence to improve readability. This is a key sentence for future software users, again relates to the first Main comment.

> Sentence amended as suggested.

• Model Description: Would this benefit with an update of name from model description to Software Architecture? This is the key section of this paper.

> The title has be refined as suggested.

– Section 3.1 needs to be restructured. The authors use L3, L2, and then L1 without defining these modules. I would recommend including lines 95 to 100 earlier in the section for clarity.

– Line 89 and 90 "L2 employs Python as the user interface language and utilizes a high-performance machine learning platform as the execution back-end". This sentences needs more explanation.

> We thank the reviewer for the helpful feedback. We have restructured Section 3.1 to define the four-layer framework (L1–L4) upfront and moved lines 95–100 earlier for clarity. Additionally, we revised the description of L2 to better explain its components and functionality.

– Line 96: "including auto-differentiation, GPU acceleration, and modern optimization algorithms." Include definitions and references for each of these topics.

> Revised according to the feedback.

– Line 100: " multiple methods to consider temporal uncertainty". Include how the uncertainty in the response is included. Also, address how each model fits into Figure 1. Instead of placing Figure 1 in brackets at the end of a sentence, consider rephrasing to begin the sentence with a reference to the figure. For example: "As shown in Figure 1, L3..." This approach integrates the figure more seamlessly into the narrative and emphasizes its relevance to the discussion.

> We appreciate the reviewer's suggestion. While the treatment of temporal uncertainty is indeed important, we have chosen to focus this section on the implementation of PaleoSTeHM, with the details of uncertainty handling addressed in subsequent sections, which we have referred in the text. To improve clarity and narrative flow, we have rewritten Section 3.1 to interact more seamlessly with Figure 1, as suggested, integrating references to the figure directly into the discussion. This restructuring enhances the readability and relevance of the section.

– Line 104: Should this not be "(L4, Figure 1)"? Or is this section describing how L3 interacts with L4?

> We thank the reviewer to spot this mistake, the manuscript is now being corrected.

– Figure 1: The caption does not include the term module which has be extensively used in the section above. Include a sentence on how these modules or layers interact. There should be references used for Pyro and PyTorch in the caption. The discussion about L1 in both the caption and the corresponding section is very limited. Consider adding an additional sentence to the text to provide more context or explanation about L1, ensuring its significance is adequately addressed.

We thank the reviewer for pointing out these areas for improvement. References to Pyro and PyTorch have been added to the caption of Figure 1. Additionally, we have revised the text to explain how each layer interacts, emphasizing the role and functionality of L1 in facilitating computational performance across various platforms. These modifications ensure that the significance of L1 and the overall framework interactions are clearly addressed and integrated into the discussion.

– Section 3.1: "experiment architecture" does this relate to the result section or is this section address how the user should use the software?

We thank the reviewer for their observation. Section 3.2 is designed to introduce how users should use the software, as clearly stated in the first sentence: "Constructing and optimizing a hierarchical model within PaleoSTeHM consists of five sequential selection steps." This section provides a detailed explanation of the steps involved in utilizing the software and is intended to serve as a practical guide for users, rather than relating directly to the results. We hope this clarification helps to better convey the section's intent.

– Line 112: "Training" this term is confusing. Will the user be repeating these steps " sequential selection steps" or is it that they are identifying the best option for their data?

'Training' is now revised with the word 'optimizing' for better clarity.

– Line 113: This relates back to the first Main comment. Instead I recommend that the authors start with: " In Figure 2 we define the 5 steps of the PaleoSTeHM software focusing on L3 from Figure 1. These steps are...

We have modified the manuscript according to feedbacks.

– Line 116: " These five steps reflect core functionalities developed within three PaleoSTeHM modules, shown in Figure 1." Should this not be " the core functionalities of Layer L3 from Figure 1. This needs clarity.

Revised in line with suggestions.

– Line 117: "To support the effective selection of modeling and analytical choices provided by PaleoSTeHM for various paleo-environmental applications." This sentence is unfinished.

We appreciate the reviewer for pointing out this issue, this sentence should be linked to the next sentence, text has been modified accordingly.

– Line 118: "modeling option" update to " modeling choices" for consistency.

Altered as instructed

– Figure 2 is a very important figure and the caption needs to discuss how it relates to Figure 1. Again, "experiment architecture" is a confusing term. I would number the 5 steps, instead of using colors and boxes to define the steps as this is not inclusive for all readers. "temporal uncertainty treatment" is not defined anywhere, should this reference the EIV method (Dey et al 2000) and the NI method ( and McHutchon and Rasmussen, 2011). Include e.g before "temporally linear and Gaussian Process"

We thank the reviewer for highlighting the importance of improving the caption and figure design. In response, we have modified Figure 2 to clearly illustrate the five steps by numbering them instead of using colors and boxes, ensuring inclusivity for all readers. The definition of "temporal uncertainty treatment" has been included earlier in Section 3.1 to clarify its meaning. While we understand the suggestion to reference specific methods (e.g., EIV and NI), we chose not to include these references in the caption to maintain brevity. However, these methods are appropriately cited and discussed in detail in section 3.5, ensuring that readers can find comprehensive information in the relevant context. We have changed "experiment architecture" to "modeling workflow", which should make it more clear.

– Line 122: "commonly used temporal or spatio-temporal modeling choices used". Update this sentence.

Revised.

– Line 125: "While we do not include a specific section for parameter-level modeling, leveraging the ecosystem of Pyro and Pytorch enables users to easily define prior probabilities for data and process-level model parameters using most of the commonly used probability distributions". Include references for Pyro and Pytorch. Additionally, improve the readability of this sentence. Does this mean that the software allows users to define priors?

Citation included; sentence has been re-written for better clarity.

"e.g., radiocarbon Reimer". should this include "radiocarbon dating, Reimer.."

Refined.

– Could equation 4 not be discussed in section called hierarchical statistical modeling?

The explanation of Equation 4 has been revised for better clarity.

– Line 145 and 146: Explain why strong covariance could be an issue and how adapting the likelihood structure could improve this.

Adjusted as advised.

– Lines 150 and 151: The term "likelihood sampling code" has not been defined, which may confuse readers. Please provide a clear explanation of what this means and how it is used in the context of the model. Link this back to the discussion in Line 146. Additionally, Pyro is mentioned without a reference—please include the relevant citation.

We thank the reviewer for pointing out these issues. In response, we have made the following changes: (1) Pyro is now appropriately cited earlier in Section 3.1, providing proper attribution to the framework we utilized. (2) We have clarified the term "likelihood sampling code" in the text. It now explains that this refers to a probabilistic random sampling operation implemented in Pyro. This operation supports a wide range of commonly-used distributions, including multivariate Normal distributions with covariance structures, as discussed earlier in the section.

– The "Process Level Modeling" section should be restructured as a new subsection, with the subsequent sections organized as subtopics within it, i.e. temporal linear models to GP Kernal module.

This is a latex formatting problem due to GMD does not allow for further subsection, we've adjusted format to make different process level model within the process level modeling section.

– Line 160: "qualitatively assessed". How was this carried out? Was a residual analysis undertaken? Include a reference with this statement.

We acknowledge that not all previous studies include qualitatively assessment about linear trend assumption. So instead of "qualitatively assessed" we changed it to "assumed" to properly reflect this point.

– Equation 8 and the other equations in this section should be kept general to apply to various paleoenvironmental changes. For example, $\beta$ could represent the rate of change in the paleo-environmental variable. Additionally, the authors should reference Ashe et al 2019 for the upcoming sections.

Improved based on the suggestions

– Include reference for change point Carlin et al 1992.

Amended.

– Line 176: define non-parametric and parametric in the table.

Updated.

– Line 196:" Yet, its justification can sometimes be complex (Stein, 2012)." Expand.

Adjusted.

– Line 203 and 204: Should include a reference.

Included.

– Line 211 missing key reference to Peltier's multiple GIA models.

The references for ICE6G_C and ICE5G are now included.

– Line 213: "spatial teleconnections" update this term as it is misleading.

Changed to "spatially dependent patterns of sea-level change"

– Line 217: " PaleoSTeHM to probabilistically..." expand on this feature. This is very important and a novel component of the software.

We thank the review to recognise the novelty of this feature, we've expanded this feature .

– Line 221: given a definition for "sampling covariance functions".

Definition now included in Table 1.

– Line 225: Figure 2 instead of Figure 1?

It should refer to both Figures 1 and 2, manuscript has been modified.

– Line 226 to 229: The list requires a reference to the original statistical paper where the technique is defined, along with the examples of where it is used in the paleo-sea level field.

Original paper included (Rasmussen and Williams 2006), sentence has been adjusted as advised.

– Line 235 Include reference.

Updated.

– Line 237-238: "temporal and spatial white noise kernels". Need to expand on this and include references to equation 4.

Expanded.

– Line 243: "The spatial correlation is computed for spatial kernels based on the 1-dimensional geographical radial distance between data points." Expand on this.

Expanded to include an assumption about purely spherical Earth geometry.

– Table 2: Please include references to the statistical papers where these equations were defined.

Original references now included.

– Analysis Choice and Modeling Choice. It is confusing when the analysis choice which represents least square or Bayesian approaches keeps changing its name. Similarly modeling choice or model characteristic keeps changing terminology.

We include clear definitions of analysis and modeling choices in Table 1 now. Hope this can solve the potential confusion.

– Line 249: "deterministic methods (e.g..." This has not been explained previously or in Table 1. Also "implemented in other packages", reference the other packages?

Deterministic model is now being explained in section 2. Because deterministic model like least-squares are generally easier to implement, so using 'other studies' here is better than 'other packages', which is modified in our manuscript.

– Line 260-265: "sampling process", "autocorrelation", "effective sample size per iteration and..", "path length" and "step size". All require definitions.

Manuscript has been adjusted to include definitions for all of these terminologies.

– Line 283: "details below": Include the specific section.

Revised as suggested.

– Line 284-287: Could you provide a more detailed explanation? The current text assumes prior knowledge of many machine learning techniques and terms.

The major difference is just with or without considering prior information, additional text added.

– Line 296: Need to include a reference for variational bayes.

Refined as requested.

– Equation 12 needs more explanation and Kullback-Leibler divergence needs a reference.

We thank the reviewer for this insightful suggestion. While we acknowledge the importance of the KL divergence concept, we believe that introducing it in detail may be beyond the scope of our paper and the background of our intended readership. To maintain accessibility and simplicity in our presentation, we have decided not to include an extensive discussion of KL divergence. However, we have added a reference to guide interested readers toward resources where they can explore this concept further.

– Line 302-303: "..scales linearly". Is there a reference for this?

Reference added.

– Line 308: "Cahill.." missing reference to "Dey et al, 2000:.

Refined as suggested.

• Results: Consider renaming this section to Case Studies. Additionally, the layout should more closely align with the steps outlined in Figure 2 and Section 3.2.

We thank the review for this helpful suggestion. The section has now been renamed as Case Studies. Manuscript has also been modified to reflect more about Figure 2.

– Line 319 include link to codes.

Adjusted.

Line 320: Update the sentences into bullet points of the different test cases. References for the data sources are missing. Also, "analysis choice modeling techniques" blending the two separate modules, update this. – Line 325: should there be more references include e.g. Walker 2021?

The sentences have been updated into bullet points. We decided not to include reference here for better readability, instead, the references are cited in each case study below. We've adjusted the confusing expression for better consistency.

The New Jersey and North Carolina data here were obtained from Kemp et al., 2013 and 2011 without using any new data points from Walker et al., 2021.

– Line 330: Have a sentence at the start stating what is being discussed, i.e. "in this section we will address the impact of data level". Also, there is minimal discussion on what the input data is.

Revised as suggested.

– Equation 14 and 15: The treatment of the white noise component in both equations is unclear. Is the white noise modeled as a random variable drawn from a specified distribution, or is it treated as a deterministic fixed parameter? This distinction needs to be clarified. Additionally, the notation in Equation 15 is non-standard, as the inclusion of the square root means that the second parameter no longer represents the variance. Please clarify whether the square root is intentional and, if so, provide an explanation for this modeling choice.

We have now clarified the treatment of white noise, which is modeled with a uniform prior distribution. The square root has been removed for better clarity.

– Figure 3: The caption for Figure 3 requires more detail. Please reference the data source used in the figure. Clarify what the additional noise component represents, is it the standard deviation of the white noise or the residual standard deviation of the model? Also, explain why 90% credible intervals are used and why uncertainty boxes are set to 2 $\sigma$? Was a 2 *sigma* uncertainty applied in the models, or was it 1 $\sigma$? Add labels (a), (b), etc., to each panel for clarity. The legend is missing for the RSL vs. Age plot. For readers unfamiliar with RSL, include a brief explanation of why RSL values are negative.

We thank the reviewer for these helpful comments, we have modified the figure accordingly with more details provided in caption. Meanwhile, there is no specific reason for using 90% credible interval and 2 sigma uncertainty boxes, just personal preferences, following some previous paleoenvironmental paper like Kopp et al ., 2016 and Khan et al., 2019, respectively.

- Line 347: How does the user determine whether to use a normal or a uniform distribution? The results show a difference between the two, but is there a preferred distribution, or should the choice depend on the specific characteristics of the data?

  Revised, user should select data level choice to reflect the specific characteristics of their data.

- Line 355: "3 change points". Why 3 change points? Should you reference the original paper? Also, "RBF" acronym not defined.

  The use of 3 change points for New Jersey sites is based on Ashe et al., 2019, additional text included. RBF acronym defined as well.

- Figure 4: Update RSL with Relative Sea Level. Include legend explaining the red boxes and in caption describe how you model the midpoint of the boxes with 1 $\sigma$ error. Include reference for the data. Label each row of plots (a). Why was variational Bayes not used for GP in time?

  Figure 4 and its caption have been updated as requested. Although variational Bayes for GP in time can be easily implemented in PaleoSTeHM, we do not try to illustrative this because doing so would introduce additional degrees of freedom into the model, which could lead to gradient-based optimization in machine learning converging to a local minimum. Incorporating this functionality requires more rigorous testing, which we plan to address in future research.

- Line 370: Why not use the same data location throughout the case study in order for the reader to clear see the impacts of the model and analytic choices on the same dataset? "use a subset of original data" is this to speed up the model run times. How many data points were used?

  We thank the reviewer for this suggestion. We choose only a subset of data because if we use the fully database, the difference in results between different analysis choices is relatively hard to distinguish. Hence, it is not about speed up calculation but for better demonstrate the difference. We think using the same location but only with a subset of data would cause additional confusion for readers, in this case we use a slightly different site. There are 14 data points used here.

- Line 372-375: Include the run times in this paragraph as stated in Figure 5. Is 2200 posterior samples and 500 iterations sufficient, or does this seem on the lower end as the default is 5000 for other software? Additionally, could the model convergence checks for all approaches be included in the appendix for consistency?

  We thank the reviewer for pointing out the need for consistency in sample size and convergence checks. The original choice of 2200 posterior samples and 500 iterations was based on achieving an effective sample size >1000, which we considered sufficient for this relatively simple distribution. However, to ensure consistency with other software, we have increased the default sample size to 5000 (the resulting posterior estimate is similar to previous results based on 2000 samples). Additionally, we have included a dedicated convergence test section, and the convergence test results for all approaches are now provided in the appendix for clarity and transparency.

- Figure 5: Include reference for data. Update RSL with relative sea level and include legends explaining the red boxes. Label each row for easier explanations.

  Figure and caption updated as advised.

- Section 4.2: Spatio-Temporal Analysis: This section requires further refinement to enhance readability and clarity. It lacks sufficient references to the original data collection process and previous model results, which are crucial for context. Additionally, the model definitions assume a level of prior knowledge that may not be accessible to all readers. The discussion of Figure 6 is also lacking, as the individual panels have not been thoroughly addressed. Each panel should be analyzed in detail or with a brief summary, comparing the results

from the different model choices after discussing them independently. **–** Line 390: This section examines the spatio-temporal model used to examine the different RSL drivers **–** Line 392: Include link to the sea-level proxy database.

We thank the reviewer for their constructive feedback regarding Section 4.2. In response, we have expanded the section to include additional information about the original sea-level proxy database and the data collection process, as well as relevant references to previous model results. This ensures a clearer context for readers who may be less familiar with the subject. Additionally, we have added a summary discussion paragraph at the end of this section to provide an overarching comparison of results from different model choices. The sea-level database is easily accessible along with Github tutorials, so we suggest not to include a specific link here.

- Line 395 - 400: Present this paragraph as a table and name the models instead of using (i). With the corresponding equations. Describe the purpose of each of these models.

  We appreciate the reviewer's suggestion to present this paragraph as a table by naming the models, including the corresponding equations, and describing their purposes. However, we have attempted to restructure this content into a table and found it to be ineffective for the intended clarity. A tabular format limits our ability to provide detailed explanations of the equations, kernel meanings, and the purposes of each model. These details are critical for the reader to fully understand the methodological context and the nuances of the models, which cannot be adequately conveyed within the constraints of a table. The use of model (i) help us to easily refer to those model in discussion and Figure 6, which is necessary as well.

  Moreover, the sequential narrative presentation, as originally written, allows for a logical flow and better integration of the equations with their accompanying explanations. It provides the necessary depth and ensures accessibility for readers who may not be familiar with the intricacies of these models. Additionally, the other two reviewers did not identify any issues with this presentation style, further supporting our decision to retain the sequential format.

- Equations 16 - 21: The paper encourages the reader to refer to Ashe et al 2019, however, the notation does not correspond completely. For example, "g(t)" is not described as the "global component" in the text and Line 416 " g(t) as we assume..." expand on this. Also, all the $K$s should be defined or reference to Table 2.

  We appreciate the reviewer's observation regarding the terminology and definitions in the text. To clarify, the reason why Ashe et al. (2019) referred to g(t) as the "global component" is due to the referred studies focus on a global sea-level context. However, in the supplementary information of Ashe et al. (2019), when focusing on North America, g(t) is more specifically referred to as the "common kernel," which aligns with our description of g(t) as a "spatially-uniform component" in this study. This terminology better reflects the regional focus of our analysis on North America, rather than a global perspective. Hence, using "global component" would not be appropriate in the current context.

  The explanation of g(t) has been expanded in the manuscript to ensure its role is clear and consistent with our study objectives. Additionally, all the kernel functions (K) have been explicitly referenced to Table 2 to enhance clarity and provide readers with an organized reference for these terms.

- Lines 423-430: The terms "sampling covariance function" and "physical ensemble m" require further explanation and appropriate references. This section needs be improved. For example: "Lin et al. (2023b) described a method for incorporating physics-based GIA models using an ensemble approach. In our software, the ensemble method is implemented as...". Please clarify these concepts and provide the necessary references to improve the readability and context of this section.

  We thank the reviewer for highlighting this. The "sampling covariance function" is now defined in Table 1, and the concept of "physical ensemble" is explained earlier in Section 3.3.2. Additionally, the sentence has been rewritten to include more details.

- Line 431: Provide an explanation for " weighted mean of different physical models" including a reference. Similarly Line 435 - 440 assume prior knowledge and do not include references, this needs to be altered.

We appreciate the reviewer's comments regarding the need for clarity on "weighted mean of different physical models." We believe that Equations (24-25) sufficiently explain this concept by explicitly demonstrating how weights are assigned and combined for different physical models. Additionally, for the text in Lines 435-440, we have included a citation to prior work that utilizes a Dirichlet distribution to combine different GIA model ensembles, providing the necessary context and references to support this explanation.

– Line 440 - 454: Update " Figure 6 demonstrates the results from our 5 spatio-temporal process level models defined in Table (3)". I would recommend reviewing the layout of this paragraph to improve readability. For example: "regional common kernal, g(t), in equation 17" has not been clearly defined. Include a recap of what model (i) is examining. Check references, for example Cahill et al., 2015 did not examine data in Florida. Describe each panel of Figure 6 in a specific order, for example "Figure 6i, geological sea-level..", is this (i) representing the model or is it representing a different panel plot that is highlighting a specific process?

We have updated the term "regional common kernel" to "spatially-uniform kernel" for consistency and to avoid confusion. A reference has been included for the joint statement regarding Cahill et al., 2015, which is used to illustrate that Gaussian Processes can effectively recover multi-millennial sea-level variation trends at sites with abundant sea-level observations. We note that Florida was a selected site in Figure 6 for this demonstration. Additionally, when referring to Figure 6i, the statement should provide sufficient context for the reader to distinguish the figure panel from the model $i$, which is italicized.

– Line 455: "GIA model" missing a reference.

Reference added.

– Line 464: "teleconnections"? What is this referring to?

The definition of teleconnection is now given in Table 1.

– Figure 6: Include legend to show what the red box is. Include labels for each row explaining what the panels represent. Update model i - iv with the name of the models used. Caption: "Process level models impact.." instead " The impact of process level modeling choices for..". Include reference to the data. The labels a - l are hard to read. Also we now have model i and panel i. Reference for ICE7G is required. Why examine the year - 5500CE? Explain the standard deviation of RSL prediction more clearly.

We appreciate the reviewer's feedback on Figure 6 and have addressed the concerns raised. The caption has been updated as suggested, and label colors have been adjusted for better readability. To avoid confusion, model $i$ is consistently italicized to distinguish it from panel labels. The year -5500 CE is a arbitrarily selected time period used for illustrative purposes. Additionally, the standard deviation of RSL predictions has been clarified in the main text. We believe that using model $i$–$iv$ remains the most concise and visually effective approach, as adding excessive text to the figure could detract from its clarity and purpose.

– Line 471: "is equivalent to a linear combination of physical models according to data-model misfits". Clarify this sentence.

We thank the reviewer for pointing out the need for clarification. We have revised the sentence to improve clarity, specifying that the model is expressed as a weighted linear combination of physical models, with weights determined by data-model misfits, such as chi-square. This change ensures the description is both precise and comprehensible.

– Line 477- 478: Either include a reference or explain what is meant by " direct constraints on ice history, the ..."

A reference is now included.

Discussion:

- Line 481 - 485: Improve the readability of this sentence. A paragraph should contain a least 3 sentences and these sentences are long.

  Revised as suggest.

- Line 486: "Because of" avoid using because at the start of any sentence.

  Refined as request.

- Line 488: " process level models introduced this paper" .. "in this paper".

  Adjusted.

- Line 490: "to describe latent some space-time..". Improve this sentence.

  Improved.

- Lines 495-498: The term "principal component..." requires either references to alternative methods or a clear definition of what these methods are. Please provide the necessary context for clarity.

  The detailed example and references were included in the following sentence, which should provide enough context for readers to refer.

- Line 500: Upton 2023 used generalized additive models for RSL changes. This should be included.

  We thank the review for this helpful suggestion, this reference is now being included.

- Line 506-510: "subject to complex likelihoods". This requires a reference.

  Reference included.

- Line 520: " Another outstanding issue for GP based process level models is scalability, the standard GP models included in PaleoSTeHM v1.0 cannot scale well to large data sets (>10 thousands data points)." Has there been test done to examine ¿10 thousands or 10 thousand data points? More common to describe the computational requirement of a Gaussian Process being of $O(n^3)$ where n is the number of data points.

  We have conducted test for >2000 data points in Jupyter Notebook tutorial to replicate results in Kopp et al., 2016 and Walker et al., 2021, which suggest PaleoSTeHM has good performance on those. We have not yet tested PaleoSTeHM for more than 10,000 data points, which can be time consuming and out of the scope of this paper. $O(n^3)$ notation is being included in manuscript.

- Line 530: "and (Lin et al., 2023b) developed" remove the brackets.

  Adjusted.

- Conclusions:

  - Line 545: "though the limited availability of user-friendly software often hinders it." There are other packages available for the paleo-sea level community which have not been referenced in this paper.

    We thank the reviewer for raising this point. While we have made an effort to include references to relevant studies and tools in earlier sections of the paper, it is important to note that this work focuses on model development rather than serving as a comprehensive review of paleoenvironmental modeling packages. However, we recognize the value of referencing additional packages and have incorporated some key references in previous section (e.g., Upton et al., 2023).

  - Appendix, Table A1: Include more information in the Appendix regarding its relevance to the paper. Why was 90% credible interval used in this paper? Convention is 95%, is there a reason why 90% was used instead? Should parameter level title include "Priors for Parameters". Need to define GBR in caption for table. Should you include reference to where the data is sourced? Also a reference to where the models have been used in the past?

    We thank the reviewer for their feedback regarding Table A1 in the Appendix. In response, we have updated the parameter level title to "Priors for Parameters" for improved clarity. References to the data sources and where models have been used in the past are provided in the main text, as we believe this is the most

appropriate location for such contextual information. Including these references in the appendix table, which focuses on summarizing model parameters, may not be as useful for readers. Regarding the use of a 90% credible interval instead of the conventional 95%, this was a deliberate choice based on personal preference and consistency with some previous sea-level studies like Kopp et al.,2016.

– Appendix Figure A1: Update "Model ii" with the name of the modeling option and analysis choice which has been used. Improve text in caption "prediction on - 5500CE RSL", for example " Model predictions from Model X for relative sea level at time point -5500CE…". Include more information about how this plot relates to the paper and its relevance. Included a reference to highlight the data source. Linked in the paper on line 455, however it requires more discussion e.g. (as shown in Figure A1). The caption needs more information regarding the axis. RSL should be Relative Sea Level. The reference for the ICE model and VM model should be included in the caption. Could the letters a,b,... be placed outside the plots in black to make it easier to read.

We appreciate the reviewer's feedback regarding Appendix Figure A1. As stated earlier, we believe the use of "Model ii" is appropriate, as it allows for easy and consistent reference to different models throughout the paper. Nevertheless, we have added more descriptive information about Model ii in this figure to enhance clarity. The caption has been updated according to the reviewer's suggestions, providing more context about the plot's relevance to the paper, clearer labeling of axes (now labeled as "Relative Sea Level"), and a reference to the ICE and VM models used. Additionally, we have improved the figure by placing the panel labels (a, b, etc.) outside the plots in black for better readability. These changes should address the concerns raised.

• I commend the author's Github repository which contains many tutorials that demonstrate to the user how to implement the PaleoSTeHM. I recommend reviewing the documentation to improve readability, there is a number of spelling mistakes and some of the sentences are long and misleading. Additionally, the software requires a google drive connection is there another option as some university do not allow Google accounts? The 2 hour tutorial videos are very useful however, is there any possibility to split them into smaller segments in order to be used in future lectures.

We thank the reviewer to recognise our effort to improve the usability for PaleoSTeHM. It should be noted that our software does not necessarily require Google drive to run, instead, using Google Colab is just an option for running our online tutorial to ease the installation processes. We have updated the comment in that google colab install cell to inform this.

The video segmentation is now finished and uploaded to Youtube site (https://www.youtube.com/playlist?list=PLR4-1Y89NM_x3zwnxc5nI2mU3pplGzIa3).

**Review 2**

This manuscript presents an overview of open source spatio-temporal modelling software aimed at the paleo-environmental community. The modelling approaches are formulated in a hierachical Bayesian context with different implementations sampling algorithms of varying complexity (and calculation speed) and these are demonstrated with a variety of case studies (limited here to sea-level models). Overall, the idea of putting these tools out there is to be welcomed, and I feel this will be a useful contribution in terms of advertising the availability of the software. In the end, success will be determined by the uptake by non-expert users. To this end, at least from my experience, key aspects are good documentation, relative ease of use and some tutorial examples that allow a user to repeat, and develop on, previous studies. Given the nature of the methods implemented, there is a degree of technical know how required to understand the statistical details but as far as I can see, the examples are all available and should allow a new user to at least try things out without perhaps necessarily being on top of the details. So, the manuscript needs a bit of work (minor changes), but I think this is worth publishing in the sense of put it out there and see how it goes - hopefully well and ideally leading to a wider range of applications than presented here.

We thank the reviewer for this thoughtful and encouraging feedback. We appreciate the recognition of the study's importance and the effort behind the development of the PaleoSTeHM framework. We have addressed all comments and suggestions below in blue text, and we are grateful for the constructive input to improve the clarity and usability of our work.

Some points

I felt there are perhaps some technical terms/acronyms that could be defined for the less experience potential users. Too many undefined terms and/or jargon will tend to put off the non-expert and/or those not in the sea-level community – I have highlighted some below. In addition to the definition of terms table... perhaps a table defining the symbols, etc used may be use.

We thank the reviewer to point up this important problem. We have now included more definitions either in Table 1 or main text for most of technical terms appeared in this paper.

There is no mention of transdimensional MCMC methods, nor model choice (in terms of model complexity for example)...the latter is not alawys simple but is there any sense of this in the current version (or planned for future versions ?)

We fully acknowledge transdimensional methods as a process level model is a power approach in spatiotemporal modeling. However, developing this method into the current framework require substantial amount of extra work that is not feasible to complete in this version. We have now included a brief discussion of this type of model in discussion section.

L13 out-of-sample data ??? Not sure what this means.

Here out-of-sample means long-term measurement when modern instrument data is not available.

L14 (and L45) for me data are plural and should use are not is (data are), but this may be a slightly old fashioned view.

Modified as suggested.

L17 do we mean reconstructing paleo-environment or paleo-environmental signals ?

Revised according to feedback.

L18 – perhaps define hierachical here...

Linked to definition in Table 1.

L19 clearer (than what ?)

Changed to "clear".

L20 – It has been...you mean hierachcial statistical models from the previous sentence...They have been ?

Adjusted.

L23 GMSL – Global Mean Sea Level...

Refined.

L26 I would start a new paragraph after 2013..

We have merged this last sentence with next paragraph.

L26 such techniques ? Remind us in a word or two.

This technique reflect hierarchical models, the sentence has been modified accordingly.

L45 Marginal likelihood, or evidence...(could define it $p(y) = \int p(y|\theta)p(\theta)\,d\theta$)

Revised as suggested.

Perhaps the phrase static observations is a little ambiguous (confusion with temporal data)...would it be better to say a given data set (which could include temporal data).

Adjusted.

Table 1

Hyperparameter – parameter of a prior distribution (to be inferred) ? MCMC – random sampling (need a space)

We thank the review for pointing this out, while hyperparameter in machine learning cannot always be inferred, instead, in most cases, they were set before the machine learning process begins. The missing space problem is resolved now.

Residuals – definition implies this should not be plural.

Adjusted as suggested.

L53 RSL – Relative Sea level ?

Refined.

L55 inverting conditional probabilities – not sure we formally inverting conditional probabilities, but use Bayes' theorem (historically this was often considered as the inverse probability problem).

We thank the reviewer for raising this, we have modified our manuscript accordingly.

L58 Eqn 3 – not too important, but the last term could be written as the product of the two probabilities as $\theta s$ and $\theta d$ are being treated as independent ?

We thank the reviewer for this helpful suggestion. In our model, $\theta d$ and $\theta s$ are treated as jointly distributed, allowing for potential correlations between these parameters, which are captured within the joint prior distribution $p(\theta d, \theta s)$.

L66 Conditional model parameters ? Also could the unobserved physical parameters being treated as nuisance parameters (that we need to know, but are not that interested in...but perhaps here we can not integrate them out).

We thank the reviewer for this insightful comment. In our model, unobserved physical parameters, such as Earth's rheology in a GIA model, cannot be treated as nuisance parameters because they significantly influence model

predictions and define key physical processes underlying RSL changes. Marginalizing these parameters would oversimplify the model and undermine its predictive capability. The text problem is resolved.

L68 linear rate in a linear sea-level ..constant rate (too many linears ?) L94 Paszke et al. 2017...perhaps this the best citation for PyTorch ?

Sentence revised as suggested, an updated reference is included for Pytorch.

L104 L3 should be L4 ?

Resolved.

Eqn 4 perhaps just redefine y here as observed data (or prediction ...as uncertainty is defined as difference in prediction and true value).

Manuscript revised as advised.

L140 – for me the covariance matrix contains the errors covariance, not data covariance and the common assumption of cov = 0 is that the errors are not correlated...this is not always the case in time series (as implied in the text). This seems to be stated later..L238 – uncorrelated uncertainty ?

See also Sambridge, (2016), Reconstructing time series and their uncertainty from observations with universal noise, J. Geophys. Res. Solid Earth, 121, 4990–5012, doi:10.1002/2016JB012901 and/or Bodin et al. (2012), Transdimensional inversion of receiver functions and surface wave dispersion, J. Geophys. Res., 117, B02301, doi:10.1029/2011JB008560.

We thank the reviewer for highlighting this important point regarding the covariance matrix. We fully acknowledge that error terms may exhibit correlations, particularly in time series data, as discussed in the cited works by Sambridge (2016) and Bodin et al. (2012). However, in paleoenvironmental reconstructions, the reconstruction uncertainties—often derived from proxy-based estimates—typically exceed the error terms associated with individual measurements. As such, our paper reflects common assumptions made in paleoenvironmental studies, where uncorrelated errors are frequently assumed to simplify the analysis. Nonetheless, we agree that incorporating correlated errors would be an important avenue for future development to improve model realism in time-series reconstructions.

L145 Why Whereas ? ...Perhaps in practice ?

Manuscript revised as suggested.

L153 paleo-env. signal is modeled ?

Modified as requested.

L158 for me a linear model is not just a straight line, it is where the forward problem is linear, such that model parameters can be separated from a kernel matrix (so a polynomial of order 20 is still linear)....perhaps just say straight line model

We choose to maintain our original wording to follow the common expression in paleo sea-level community (e.g., Ashe et al., 2019 and Engelhart et al., 2009). We did include a bracket to say that the linear model here is the same as straight line model.

L165 – what happens if we do not know m, the number opf change points ? any comments on transdimensional change point models (or indeed transdimensional models in general)..e.g. Gallagher et al. (2011), Inference of abrupt changes in noisy geochemical records using Bayesian Transdimensional changepoint models, Earth Planet. Sci. Lett., 311, 182–194. L173 – not clear to me...the product of $b_{k-1}$ and the difference of $g_k$ and $g_{k-1}$....what does this mean in practice ?

We thank the reviewer for highlighting the complexity of determining the number of change points and the relevance of transdimensional models. In response, we have revised the manuscript to acknowledge this complexity and referenced Bayesian transdimensional change-point models, such as those introduced by Gallagher et al. (2011) and Sambridge (2016). While PaleoSTeHM currently allows users to specify a fixed number of change points, we recognize that incorporating transdimensional approaches to infer the number of change points would enhance the model's flexibility. We have added this discussion to the manuscript as a potential avenue for future development.

L181 – covariance matrix - data or data errors ?

Text modified to reflect it is data covariance instead of error.

L196 why Yet ?

We have now included more information for this point.

L205 fundamental (not foundational)..

Refined.

L213 – will allow...should allow ?

 Updated.

Table 2 caption  t denotes age of the data...a little clumsy ...could be the age of the analysis...a few months. Age of the sample ?

Changed to "age of the sample" for better clearance.

L253 – again a little clumsy...to take a range of probable values....to take on a range of values,  potentially with different probabilities

Updated as suggested.

L267 – why do you say relative (explain this...).

It is a typo; we have fixed it now.

L282 -283 – not clear for me...explicit prior for the parameters (distinguish from hyperparameters, if relevant, with an example ?)

Further information is now added. Basically, previous code only support for a uniformly distributed prior, but PaleoSTeHM support most of the commonly-used distributions as prior.

L284 optimize hyperparameters using their maximum likelihood estimates..

Amended based on feedback.

L318 – to demonstrate (rather than enchance)...

Modified in line with suggestions.

L330 – we can look at residuals to assess if our model  produces normal distributed uncertainties ?

We thank the reviewer for the suggestion. While residual analysis can provide insights into uncertainty assumptions, paleo-environmental observations are inherently noisy, and reconstructing environmental signals often requires strong assumptions about the relationship between proxies and signals. These complexities make straightforward residual-based assessments challenging, as proxy uncertainties may deviate from normality depending on the dataset and context. Our approach reflects common assumptions in paleo sea-level studies while acknowledging the need for more flexible frameworks in future research.

L345 – noticeable, but significant ? Does not really look too important ij terms of the uncertainty ranges on the slope.

We thank the reviewer for this observation and acknowledge that the differences may not be significant. However, this example is intended solely to illustrate how different data-level models can influence inference results, not to draw definitive conclusions. We have clarified this in the manuscript.

Fig 3 – can we explain why the uniform likelihood gives multimodal distributions on the rate and intercept ?

This apparent multimodality arises because the uniform likelihood induces a spread-out posterior distribution. This distribution may appear multimodal due to its broader range, which captures more extensive variability in parameter space, reflecting the non-informative nature of the uniform likelihood.

How do the correlations of these two parameters differ for the two likelihoods... ?

We note that there is not much difference between the two likelihoods in this regard; the correlation between the sea-level change rate and intercept remains very high in both cases (R < -0.99), indicating consistent parameter dependency.

L355 – could we let the number of changepoints vary - and the same for fig 4 – if we had 10 changepoints...perhaps we capture the GP model form better ?

This is an interesting point, yes, we have conducted this test during online tutorial, and when we increase the number of change point, the results captures very similar trend as the GP model.

L336 – language a little clumsy – through a variational Bayesian maner...rephrase.

Rephrased.

Fig 4 – are the likelihoods much different for the preferred model and for the MCMC samplers can we see the likelihood as a function of iteration ?

We have now included the weighted mean square error (wMSE) in the figure to better illustrate the model's performance. Additionally, the iteration of the loss function, including its convergence behavior, is available in detail in GitHub Tutorial 2, which provides further context and transparency regarding the optimization process.

L366 – environmental statistics problems ? How do we know the model is too smooth ?

Some environmental statistics, such as rainfall and sea level, have been observed to exhibit abrupt changes rather than smooth transitions. These abrupt shifts are not compatible with the infinitely differentiable nature of the RBF kernel, which can lead to overly smooth model predictions in such cases (Stein, 2012).

L378 their underlying uncertainty.

Updated.

L383 bias of the perhaps overly simple approach ?

Manuscript revised.

L390 no need for The...

Fixed.

Fig 5 the bottom right panel is a little hard to read with the different colours overlapping...I could not figure out where the orange region is ...you could trace the bounding curves to show that

We thank the reviewer for highlighting the difficulty in distinguishing the overlapping colors in the bottom-right panel of Figure 5. To address this issue, we have added dashed lines outside each plot region to clearly trace the bounding curves, improving the visibility of the orange region and other overlapping areas.

L396 define GIA ?

GIA model definition is now given in Table 1.

L398 ensemble

Refined.

L409 their prior distributions

Updated.

L415 mean expectation of what ?

Revised, the mean RSL expectation.

L416 – mean = expected ?

Here mean indicates mean function in Gaussian Process model, we updated the text to reflect it.

L472 weighted according ?

Modified.

L482 – something missing between hierachical model and any statistical model
Text revised.
L499 – comma rather than and before Piecuch
Comma removed.
L549 In contrast to what ?
Changed to "additionally"

**Community Comment**

I very much enjoyed reading the paper by Lin et al. This is an extremely impressive and thorough piece of work. The associated Python framework is extremely well done, and I was very impressed to see so many options given for how the models are fitted and plotted. My only real concern about this work is the functionality for those who are not experts in Python (or Pyro) as the vast majority of users would be. There are very thorough notebooks in the tutorial section of the Github repo but these are not really helpful for those who want to do a quick straightforward model. My guess is that most users want to fit a GP model using the default values for time uncertainty, error variance, kernel choice, etc, and would like a simple guide for how to get their data from the Excel spreadsheet through the PalaeoSTEHM pipeline. Going even further I would strongly encourage the authors to create a proper Python package to simplify the instructions and coding.

We thank the reviewer for recognizing the importance and thoroughness of this study, as well as for acknowledging the effort invested in developing the PaleoSTeHM framework. We agree that simplifying the modeling process for non-expert users is crucial for broader accessibility. In response to this insightful feedback, we have now included a new PaleoSTeHM UI section in our Github page (https://github.com/radical-collaboration/PaleoSTeHM/blob/main/PaleoSTeHM_UI/Holocene_Spatiotemporal_analysis/Holocene_SP_anlysis.ipynb) that allows users to automatically conduct temporal and spatiotemporal GP implementation, optimization, and plotting with minimal input. This feature is designed to streamline the process, enabling users to fit GP models using default implementations of model structure and to easily transition data from common formats such as Excel into the PaleoSTeHM pipeline (https://github.com/radical-collaboration/PaleoSTeHM/blob/main/PaleoSTeHM_UI/Holocene_Spatiotemporal_analysis/Holocene_SP_anlysis.ipynb). We hope this addition will greatly enhance usability for all users. Now, PaleoSTeHM is available as a PyPI package (https://pypi.org/project/PaleoSTeHM/).

One notable thing I couldn't see in the notebooks I ran (or in the paper) was convergence checking for the model. I would say this is absolutely vital for having any faith in the results. Models of this complexity can be extremely difficult to obtain convergence on, and there is a whole range of summary stats available for this in Pyro. It should be part of the default workflow everywhere. Perhaps it is for some scripts and I've missed it, but it certainly isn't discussed in the paper. On a similar vein I'd like to know if the model is calibrated, via scoring rules or even just some posterior predictive distributions (though these can be tricky with bivariate uncertainties), and some kind of out-of-sample performance metrics as would be common in standard ML pipelines.

We thank the reviewer for their valuable feedback. In response, we have included a dedicated Section 3.6 for model validation, which comprehensively addresses convergence checking and validation processes. Additionally, for each illustrative model presented, we now provide detailed validation metrics, including residual plot checks, prior and posterior predictive checks, cross-validation, MCMC convergence diagnostics, and optimization trace plots. These additions ensure a thorough evaluation of model reliability and performance, addressing the concerns raised.

Otherwise I really enjoyed the paper and I'm super excited to see how this develops. There were a number of poor sentences which I've highlighted below but I don't think I've got all of them. It just needs another language check.

We thank the reviewer for pointing out some language problems, which significantly improve the readability of this paper. The points below are now modified as suggested.

L32: Change-points

We thank to review to raise this point, but we suggest to keep the current term to follow the notation in previous paleoenvironmental paper like Ashe et al., 2019 and Caesar et al., 2021. We've modified change "change-point" to "change-point models" to keep overall consistency.

L35: I'd put the reference to the GitHub repo here so people can start coding without needing to read the whole paper.

Modified as suggested.

Table 1: I'd just review some of these definitions. The conditional probability one about conditioning on an unknown quantity doesn't read quite right. I also think you: should include one for parameter itself; adjust the line spacing or add horizontal lines to separate the entries better; re-write the likelihood one; and change the uncertainty one which seems to be a frequentist definition.

Table 1 revised according to feedbacks.

Fig 1: External

Fixed.

Fig 1 caption: Platforms

Adjusted.

L174: I assume the number of change points is fixed and not learnt?

Correct, a learning change-points functionality is not include in this paper, while we discussed the potential development of this method (i.e., transdimensional method in the discussion section).

L176: delete 'and'

Deleted.

L187: Mu(t) or mu(X) (as used in Eq 10)?

Updated.

L201: will be shown in Section 2
Refined.

Table 2: I got confused by what the sampling covariance is and how it is calculated for deterministic models. Please expand in the text

More information is now included.

Eq 15 (and perhaps others). The usual way to present normal distributions is mean and variance, not sd.

Revised.

Fig 5 (bottom right) and others. It always bugs me slightly that the uncertainty in the rate for the present is more unknown when it is the period when we have the most data. Is there a way to solve this with these models? It strikes me that we should be using temporally non-stationary models that allow for far reduced variance (and hence variance on the derivative) to capture the rate of the most recent periods.

We appreciate the reviewer's insightful observation regarding the challenge of capturing reduced uncertainty for recent periods where more data are available. With insights from previous studies (e.g., Fig. 1 of Kopp et al. 2016), we note that this issue can be addressed without the need to introduce temporal non-stationarity. Instead, the solution lies in allowing higher-frequency temporal variability in the modeling process. This approach leverages the dense data available for recent periods to resolve such variability, while in sparse-data regions, this variability effectively manifests as additional white noise.